# Motion sensing superpixels (MOSES) is a systematic computational framework to quantify and discover cellular motion phenotypes

Felix Y Zhou[1][†]*, Carlos Ruiz-Puig[1][†], Richard P Owen[1], Michael J White[1], Jens Rittscher[1,2,3]*, Xin Lu[1]*

[1]Ludwig Institute for Cancer Research, Nuffield Department of Clinical Medicine, University of Oxford, Oxford, United Kingdom; [2]Institute of Biomedical Engineering, Department of Engineering, University of Oxford, Oxford, United Kingdom; [3]Big Data Institute, Li Ka Shing Centre for Health Information and Discovery, University of Oxford, Oxford, United Kingdom

**Abstract** Correct cell/cell interactions and motion dynamics are fundamental in tissue homeostasis, and defects in these cellular processes cause diseases. Therefore, there is strong interest in identifying factors, including drug candidates that affect cell/cell interactions and motion dynamics. However, existing quantitative tools for systematically interrogating complex motion phenotypes in timelapse datasets are limited. We present Motion Sensing Superpixels (MOSES), a computational framework that measures and characterises biological motion with a unique superpixel 'mesh' formulation. Using published datasets, MOSES demonstrates single-cell tracking capability and more advanced population quantification than Particle Image Velocimetry approaches. From > 190 co-culture videos, MOSES motion-mapped the interactions between human esophageal squamous epithelial and columnar cells mimicking the esophageal squamous-columnar junction, a site where Barrett's esophagus and esophageal adenocarcinoma often arise clinically. MOSES is a powerful tool that will facilitate unbiased, systematic analysis of cellular dynamics from high-content time-lapse imaging screens with little prior knowledge and few assumptions.
DOI: https://doi.org/10.7554/eLife.40162.001

**\*For correspondence:**
felix.zhou@ludwig.ox.ac.uk (FYZ);
jens.rittscher@eng.ox.ac.uk (JR);
xin.lu@ludwig.ox.ac.uk (XL)

[†]These authors contributed equally to this work

## Introduction

During tissue development and homeostasis in multicellular organisms, different cell types expand and migrate to form defined organ structures. For example, during wound-healing, both immune and epithelial cells are required to proliferate and coordinately migrate (*Schaffer and Nanney, 1996*; *Clark, 2013*; *Leoni et al., 2015*). Aberrant cellular motion can be caused by deregulation of key signalling pathways in pathological conditions, including cancer, and may be critical for disease development and progression, for example leading to invasion and metastasis. Therefore, there is strong biological and clinical need for precise, quantitative characterisation of cellular motion behaviour in a tissue-relevant context to objectively compare the effects of drugs and genetic mutations.

When two cell populations adjoin *in vivo*, they often form a sharp, stable interface termed a 'boundary', with limited intermingling (*Dahmann et al., 2011*). In adult humans, sharp boundaries separate different types of epithelia, such as between the squamous and columnar epithelia in the esophagus, cervix and anus. Disruption of these boundaries can lead to disease. For example, disruption of the esophageal squamous columnar epithelial boundary is a feature of Barrett's

Esophagus (BE), a condition that confers a 30–50 fold increased risk of esophageal adenocarcinoma (EAC) (*Gaddam et al., 2013*). Understanding how complex tissue motion dynamics relate to pathological phenotypes, and how they can be affected by intrinsic and extrinsic factors, is therefore a key challenge in biomedical research. Live cell imaging over a long time (e.g. hours/days) is required to study the underlying complex cellular motion.

Recently, there has been a rising interest in scaling up live cell imaging for unbiased high-throughput screening and analysis to identify drug targets and develop therapies that can cause or prevent abnormal cellular motion (*Schmitz et al., 2010*; *Held et al., 2010*; *Pau et al., 2013*). Accordingly, a diverse range of co-culture experimental assays have been developed (*Goers et al., 2014*), amongst which wound-healing type assays are one of the simplest and most widely used. Attempts have been made to use these assays to study the regulation of complex biological/pathological processes such as the stable boundary formation between homo- and heterotypic cell populations using live cell imaging over a period of up to 6 days. However, a major barrier is how to analyse the resulting complex biological motion phenotypes and its variation between different tested conditions, particularly for multi-cell type populations (*Goers et al., 2014*) in high-content screens.

In general, there are two types of cell movements; 'single-cell' migration in which each cell migrates independently or 'collective' migration where a group of cells migrates together in a coordinated fashion. Single-cell migration has been well studied due to the availability of many single-cell tracking methods that can extract rich motion features even for unbiased high-throughput screens (*Padfield et al., 2011*; *Meijering et al., 2012*; *Maška et al., 2014*; *Schiegg et al., 2015*; *Nketia et al., 2017*). In contrast, collective migration is much less understood due to a lack of tools to extract equally rich quantitative motion features of cell populations in a high-throughput fashion with minimal prior knowledge.

Extension of existing single-cell tracking methods to analyse collective cell motion in general is non-trivial. Single-cell tracking methods all require accurate single-cell image segmentation, which is highly challenging when cells adjoin or overlap as they do frequently in tissue and confluent cell cultures. Moreover, there is a lack of systematic techniques to associate the motion of individual cells to the global moving collective. Alternatively, existing popular methods to analyse collective motion in cell populations such as Particle Image Velocimetry (PIV) (*Szabó et al., 2006*; *Petitjean et al., 2010*) and its variants (e.g. cell image velocimetry (CIV) (*Milde et al., 2012*)), do not require image segmentation. However, they only extract the global motion for a single video frame in the form of a velocity field, a velocity vector for each image pixel without assignment of pixels to individual cells. This lack of continuous tracking of cellular motion limits the application of PIV methods when characterising collective migration in complex biological phenomena such as boundary formation and chemoattraction. These biological processes often occur over a highly variable time period ranging from minutes to days, and identification of involved cell groups is often desired. Furthermore, PIV-extracted motion fields fail to exploit temporal continuity to reduce noise and avoid perturbations by imaging artefacts including visual clutter, image occlusion, and autofluorescence. Most prohibitively, PIV methods output only the velocities of image pixels. There is a lack of a systematic method to build motion 'signatures' to automatically and unbiasedly discriminate and predict biological motion phenotypes in large datasets. To this end, previous studies have attempted to extract additional motion parameters from PIV and appearance parameters from individual video frames (*Neumann et al., 2006*; *Zaritsky et al., 2012*; *Zaritsky et al., 2014*; *Zaritsky et al., 2015*; *Zaritsky et al., 2017*). These include applying velocity-based clustering in migrating monolayers, and handcrafted image features to describe local image appearance using, for example, local binary patterns (*Zaritsky et al., 2012*). However, these approaches are sensitive to the number of identified phenotypes and data outliers, requiring relatively high-quality imaging, restricting the experimental systems they can be applied to. Importantly, all the approaches (*Neumann et al., 2006*; *Zaritsky et al., 2012*; *Zaritsky et al., 2014*; *Zaritsky et al., 2015*; *Zaritsky et al., 2017*) assume prior knowledge of the motion behaviour. Notably, kymographs specifically exploit motion patterns known to be symmetrical around a given spatial axis such as wound healing and cannot be used to identify novel spatial motion features that could be relevant to normal tissue development or to a defined disease setting.

A suitable computational method for studying cellular motion therefore should be able to address the joint challenge of analysing single-cell and collective motion behaviour. Additionally, it must be: (i) scalable in a medium- or high-throughput manner; (ii) sufficiently robust to handle inevitable

variations in image acquisition and experimental protocol; (iii) sensitive to detect motion differences resulting from small changes in environment or stimuli with minimal replicates; (iv) automatic, requiring no manual intervention except for initial setting of parameters; and (v) unbiased to enable motion characterisation (e.g. as a motion 'signature') with minimal prior assumptions of motion behaviour.

To address these analytical challenges, here we developed Motion Sensing Superpixels (MOSES), a computational framework that aims to provide a flexible and general approach for biological motion extraction, characterisation and phenotyping. We empowered PIV-type methods with a mesh formulation that enables systematic measurement and unbiased extraction of rich motion features for single and collective cell motion suitable for high-throughput phenotypic screens. We use the analysis of a multi-well plate-based *in vitro* assay to study the complex cell population dynamics between different epithelial cell types from the esophageal squamous-columnar junction (SCJ) to demonstrate the potential of MOSES. Our analysis illustrates how MOSES can be used to effectively 'encode' complex dynamic patterns in the form of a motion 'signature', which would not be possible using standard globally extracted velocity-based measures from PIV. Finally, a side-by-side comparison with PIV analysis on published datasets illustrates the biological relevance and the advanced features of MOSES. In particular, MOSES can highlight novel motion phenotypes in high-content comparative biological video analysis.

## Results

### *In vitro* model to study the spatio-temporal dynamics of boundary formation between different cell populations

To develop MOSES, we chose to investigate *in vitro* the boundary formation dynamics between squamous and columnar epithelia at the esophageal squamous-columnar junction (SCJ) (*Figure 1A*). To recapitulate features of the *in vivo* boundary formation, we used three epithelial cell lines in pairwise combinations and an experimental model system with similar characteristics to wound-healing and migration assays but with additional complexity. Together the resulting videos pose a number of analytical challenges that require the development of a more advanced method beyond the current capabilities of PIV and CIV.

To model the relevant esophageal interfaces, we used three epithelial cell lines: EPC2, an immortalised squamous epithelial cell line from the normal esophagus (*Harada et al., 2003*); CP-A, an immortalised BE cell line with gastric columnar epithelial properties (*Merlo et al., 2011*); and OE33, derived from EAC (*Boonstra et al., 2010*). We co-cultured these lines in the following combinations: 1) EPC2:EPC2 (squamous:squamous, as a normal control); 2) EPC2:CP-A (squamous:columnar, as in Barrett's esophagus); and 3) EPC2:OE33 (squamous:cancer, as in EAC) (*Figure 1B*). Two epithelial cell populations, each labelled with a different lipophilic membrane dye (*Progatzky et al., 2013*), were co-cultured in the same well of a 24-well plate, separated by a divider with width 500 µm. The divider was removed after 12 h and cells were allowed to migrate towards each other (*Figure 1C*).

We first compared the effects of the two dyes on proliferation and migration using monolayer combinations of the three cell types (*Video 1*). Proliferation was assessed by cell density, automatically counted from DAPI staining using convolutional neural networks (CNN; see *Figure 1—figure supplement 1*, Materials and methods). Migration (diffusive behaviour) was assessed by mean squared displacement (*Park et al., 2015*). Green and red fluorescent-labelled EPC2, CP-A, and OE33 cells proliferated at the same rate (*Figure 1D*, for EPC2 cells: slope = 0.976, Pearson correlation coefficient 0.978, *Figure 1—figure supplement 1E*), and migrated in a similar way (*Figure 1—figure supplement 2*). Compared to non-dyed cells, the mobility of CP-A cells were unaffected however both dyes equally reduced the mobility of EPC2 and OE33 cells. The diffusion modes of all cells were unaffected (*Figure 1—figure supplement 2*) and the boundary formation behaviour between dyed and non-dyed cells was identical.

We next analysed boundary formation in the three different combinations of co-cultured epithelial cell lines (EPC2:EPC2, EPC2:CP-A, and EPC2:OE33), initially in serum-containing media (5% fetal bovine serum, FBS). In all three combinations, as expected, both populations moved as epithelial sheets (*Video 2*). Firstly, in the squamous combination using the same EPC2 cell line labelled with two different colours, the green- or red-labelled EPC2:EPC2 cells met and coalesced into a

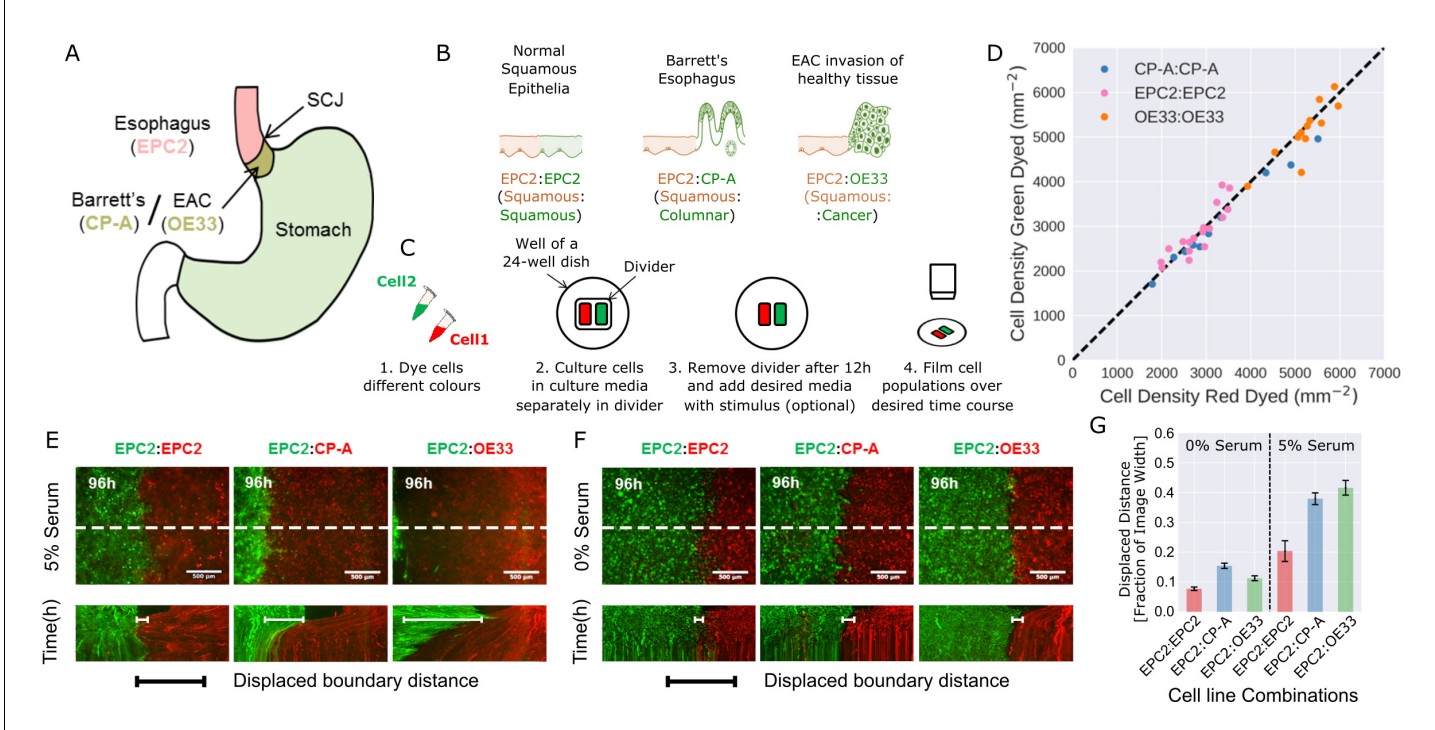

**Figure 1.** Temporary divider system to study interactions between cell populations. (**A**) The squamous-columnar junction (SCJ) divides the stratified squamous epithelia of the esophagus and the columnar epithelia of the stomach. Barrett's esophagus (BE) is characterised by squamous epithelia being replaced by columnar epithelial cells. The three cell lines derived from the indicated locations were used in the assays (EPC2, squamous esophagus epithelium, CP-A, Barrett's esophagus and OE33, esophageal adenocarcinoma (EAC) cell line). (**B**) The three main epithelial interfaces that occur in BE to EAC progression. (**C**) Overview of the experimental procedure, described in steps 1–3. In our assay, cells were allowed to migrate and were filmed for 4–6 days after removal of the divider (step 4). (**D**) Cell density of red- vs green-dyed cells in the same culture, automatically counted from confocal images taken of fixed samples at 0, 1, 2, 3, and 4 days and co-plotted on the same axes. Each point is derived from a separate image. If a point lies on the identity line (black dashed), within the image, red- and green-dyed cells have the same cell density. (**E,F**) Top images: Snapshot at 96 h of three combinations of epithelial cell types, cultured in 0% or 5% serum as indicated. Bottom images: kymographs cut through the mid-height of the videos as marked by the dashed white line. All scale bars: 500 µm. (**G**) Displaced distance of the boundary following gap closure in (**E,F**) normalised by the image width. From left to right, n = 16, 16, 16, 17, 30, 17 videos.

DOI: https://doi.org/10.7554/eLife.40162.002

The following figure supplements are available for figure 1:

**Figure supplement 1.** Automated cell counting with convolutional neural networks (CNN).
DOI: https://doi.org/10.7554/eLife.40162.003
**Figure supplement 2.** Cell migration is largely unaffected by dye colour.
DOI: https://doi.org/10.7554/eLife.40162.004
**Figure supplement 3.** Motion fields at gap closure between EPC2:EPC2, EPC2:CP-A and EPC2:OE33 cell line combinations.
DOI: https://doi.org/10.7554/eLife.40162.005
**Figure supplement 4.** Gap closure times and cell proliferation of cell line combinations in 0% and 5% serum.
DOI: https://doi.org/10.7554/eLife.40162.006
**Figure supplement 5.** Collective sheet migration dynamics are lost in 0% serum.
DOI: https://doi.org/10.7554/eLife.40162.007

monolayer as expected. Secondly, in the squamous-columnar EPC2:CP-A combination, we observed boundary formation between the two populations after 72 h, following a short period of CP-A 'pushing' EPC2. Thirdly, in the squamous-cancer EPC2:OE33 combination, the cancer cell line OE33 expanded continuously, resulting in the disappearance of EPC2 from the field of view (***Video 2***, ***Figure 1E***) as assessed by the motion field and confocal images (***Figure 1—figure supplement 3***). The forces that govern the behaviour of the two cell lines on contact are unknown and traction force microscopy is required to investigate the 'retracting' or 'pushing' behaviour of EPC2 or OE33 cells respectively in future studies. Nonetheless, the observed boundary formation in the EPC2:CP-A

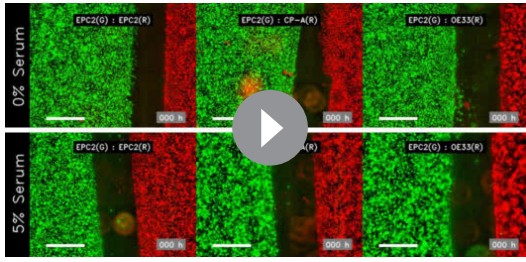

**Video 1.** Dynamics of monolayer combinations of three oesophageal epithelial cell lines, EPC2:EPC2, CP-A:CP-A, OE33:OE33. Bar: 500 μm.
DOI: https://doi.org/10.7554/eLife.40162.008

combination and disordered interactions between EPC2:OE33 cells suggest that the three chosen epithelial cell lines in the tested combinations may model certain motion dynamics of the *in vivo* cell-cell interactions. Interestingly, cell counting suggests the observed phenotypes were not due to differential proliferation rate between the individual cell lines in combination (*Figure 1—figure supplement 4C,D*).

Evidence from model systems, including *Drosophila melanogaster* embryonic parasegment (*Monier et al., 2010*) and anteroposterior and dorsoventral wing disc boundaries (*Major and Irvine, 2005*; *Major and Irvine, 2006*; *Landsberg et al., 2009*), suggests the importance of cell-cell interactions and collective migration for stable boundary formation between epithelial populations *in vivo*. Thus to create a dataset in which we predict that cell motion would be dramatically disrupted we repeated the assays in serum-free conditions. We used convolutional neural networks (Materials and methods) to determine the mean cell density and mean change in cell density from video frames over the first 48 h of cells grown in serum (5% FBS) and no serum (0% FBS) conditions. We observed that culturing of the same cell combinations in serum-free versus 5% serum medium had undetectable impact on cell density at the very confluent cell conditions investigated (*Figure 1—figure supplement 4*). However, as expected, serum-free condition induced large global changes in cell migration (*Figure 1F*, *Figure 1—figure supplement 5A*) with observed loss of cell contacts, collective sheet migration (*Figure 1—figure supplement 5B,C*) and the absence of boundary formation. We also observed reduced overall boundary displacement (*Figure 1F,G*) and all cell line combinations exhibited similar motion dynamics (*Figure 1F,G*, *Video 2*). These results illustrate that serum-free medium has a profound impact on cell motion dynamics and the generated video datasets in serum-free medium are ideal as an experimental condition for testing the ability of MOSES to detect motion dynamics under different experimental conditions. It is important to note that the serum-free condition is not used as a biological negative control of boundary formation but only as a computational negative control for the development of our method.

## Development of MOSES to quantify cell motion dynamics

We next developed a computational workflow, MOtion SEnsing Superpixels (MOSES), to quantify cellular motion from video datasets. MOSES was formulated modularly with three components (*Figure 2*): 1) motion extraction; 2) construction of long-time motion tracks; and 3) capture of local dynamic context.

**Video 2.** Sheet migration in EPC2:EPC2, EPC2:CP-A and EPC2:OE33 in 0% and 5% FBS. No boundary forms in 0% fetal bovine serum (FBS) where all combinations move similarly. A stable boundary forms between EPC2 and CP-A in 5% FBS. Film duration: 96 h. Bar: 500 μm. Videos were contrast enhanced for better visualisation.
DOI: https://doi.org/10.7554/eLife.40162.009

## Component 1: Motion extraction

We use dense optical flow to avoid the shortcomings of using cell segmentation to track individual cells in confluent tissue. Optical flow is similar to particle image velocimetry (PIV) but yields higher resolution motion fields with a displacement vector for each pixel at every time point. PIV typically estimates a single velocity for an image patch or 'window' using spatial correlation. This corresponds to extracting only the superpixel movement in *Figure 2A* over one frame. Optical flow is also easier to modify to account for additional physical phenomena such as out-of-plane motion and large, discontinuous movements not captured within a PIV 'window' (*Brox et al., 2004*; *Brox et al., 2009*).

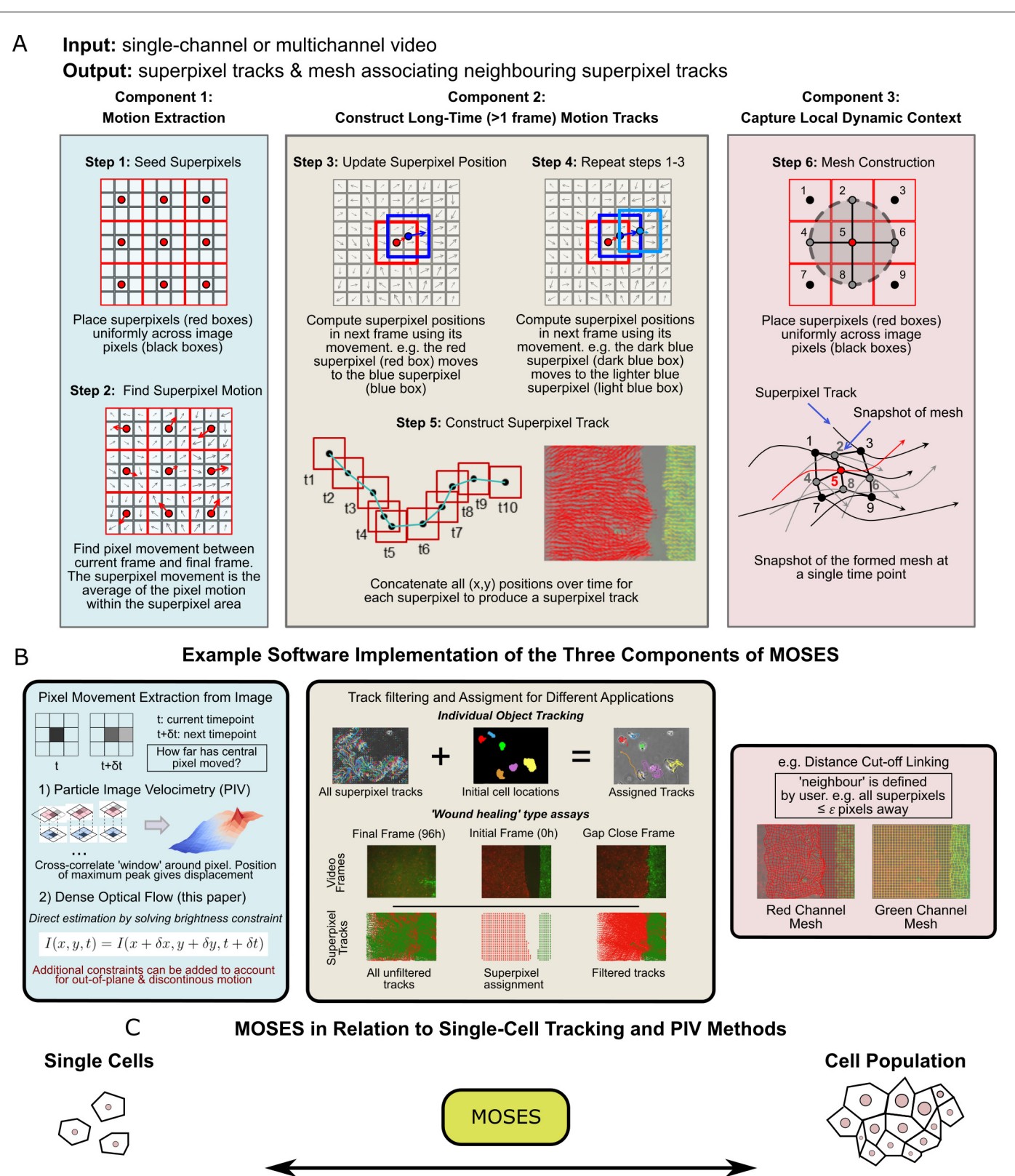

**Figure 2.** Schematic diagram of MOtion SEnsing Superpixels (MOSES). (**A**) High level overview of the three configurable components that define MOSES: motion extraction, construction of long-time motion tracks, and capture of local dynamic context. Long-time here indicates tracking of
*Figure 2 continued on next page*

*Figure 2 continued*

superpixels for longer than one frame or timepoint. (**B**) Example ways to practically implement each of the three high-level MOSES component concepts described in (**A**) in software. (**C**) Using long-time tracking of superpixels MOSES bridges single-cell tracking for sparse cell culture and PIV for confluent monolayers to extract global motion patterns for both scenarios, single-cell and collective motion within one computational framework.

DOI: https://doi.org/10.7554/eLife.40162.010

The following figure supplements are available for figure 2:

**Figure supplement 1.** Intensity-independent filtering of superpixel tracks for migrating epithelial sheets.

DOI: https://doi.org/10.7554/eLife.40162.011

**Figure supplement 2.** Experimental validation of MOSES optical flow superpixel tracking.

DOI: https://doi.org/10.7554/eLife.40162.012

## Component 2: Construct long-time motion tracks

To capture all spatial motion dynamics in the video, we construct long-time motion tracks that continuously track the cellular motion for any amount of time using superpixels. Local neighbouring image pixels are grouped into a pre-specified (see below) number of image regions or 'superpixels' to enable spatial averaging of the pixel-based motion fields (*Figure 2A*, Step 1). At each time point, the $(x,y)$ position of a given superpixel is updated by averaging the optical flow (*Farnebäck, 2003*) over the image area covered by the superpixel to obtain the displaced direction and distance (*Figure 2A*, Steps 2-4). Subsequently, long-time superpixel motion tracks are generated by concatenating the positions of each superpixel at all time-points (*Figure 2A*, Step 5). All long-time superpixel motion tracks together encode the entire spatio-temporal motion history within the video. By default, the number of superpixels used are user-specified and fixed to sufficiently sub-sample only the field-of-view of the initial video frame. Where a video exhibits highly dynamic movement and coverage of the full spatio-temporal motion, including the movement of new cells not initially present but which later move into the field of view, is necessary for analysis, additional superpixels can be dynamically added during tracking. Tracks produced by the latter approach are known as dense trajectories (*Wang et al., 2011*). In our experiments, we used a fixed number of 1000 superpixels to cover an image size of 1344 x 1024 (with this setting an average superpixel is 37 pixels x 37 pixels covering $\approx$ 14,000 $\mu m^2$ (2x lens) or $\approx$ 3600 $\mu m^2$ (4x lens)) to monitor epithelial sheet dynamics. For multichannel images, each channel representing different coloured cell populations (i.e. red and green in our assay) was independently tracked in this manner (*Figure 2B*).

Depending on the specific application, background artefacts such as stage drift or floating debris, can contribute tracks that are biologically irrelevant and need to be filtered in post-processing (*Figure 2B*, middle panel). In this filtering step, superpixels are assigned to cover only the dynamic motion of all 'objects' of interest. The 'object' for single-cell tracking is each individual cell, and for epithelial sheets is the entire sheet within the video frame. To assign superpixels to each sheet, we use motion information (*Figure 2—figure supplement 1*, Materials and methods). This approach avoids relying on image intensity features whose variation commonly leads to segmentation errors.

We validated our motion extraction and long-time track construction using published single-cell tracking datasets from the cell tracking challenge (*Maška et al., 2014*) (*Video 3*), and also experimentally, by spiking one of the dyed cell populations in the *in vitro* setup described above with a third, sparse population of blue-dyed cells. Our method produced very similar tracks (similarity score of 0.8) to an open software tool for single-particle tracking, TrackMate (*Tinevez et al., 2017*) (*Figure 2—figure supplement 2*, *Video 4*).

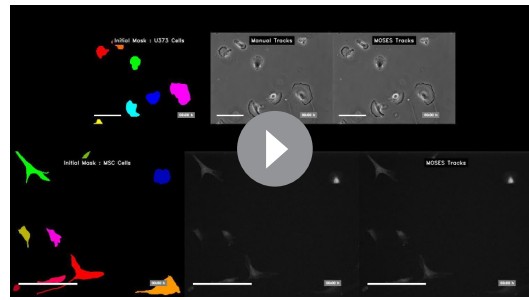

**Video 3.** Single-cell motion extraction using MOSES workflow compared to manual annotations used in the cell tracking challenge dataset (*Maška et al., 2014*). Bar: 100 μm.

DOI: https://doi.org/10.7554/eLife.40162.013

## Component 3: Capture local dynamic context

A 'mesh' is constructed by 'linking' superpixels to capture how individual superpixels move with respect to their neighbours over time (*Figure 2A*, Step 6). The constructed dynamic mesh, inspired by work in human surveillance (*Chang et al., 2011*), naturally captures the local collective dynamics without explicit thresholding or clustering. Separate meshes are produced for each colour channel. Multiple different meshes can be defined and constructed for specific purposes (see Materials and methods). Because the mesh is a representation of interactions inferred from the local spatial relationships of superpixels, it can be used to derive robust local and global measures of motion relating to biological phenomena such as collective motion and tissue interactions (see below).

In summary, MOSES has been designed to describe the video motion in terms of the individual trajectories of moving 'entities' (defined by the image superpixels) and their spatiotemporal interactions (as captured by a dynamic mesh). Depending on the application, superpixels can be assigned to and biologically interpreted as part of an object, a single cell or multiple cells.

## Quantitative measurement of squamous and columnar epithelial boundary formation using MOSES

To verify the enhanced ability of MOSES over standard PIV analysis to quantitatively assess relevant biological features of interest in tissue, we applied it to analyse the formation of boundary dynamics between squamous and columnar cells. In total, 125 videos (48 with 0% serum and 77 with 5% serum) were collected from four independent experiments and jointly analysed. These videos are highly heterogeneous, creating a challenging dataset for analysis (*Supplementary file 1*, *Figure 3—figure supplement 1*).

Standard velocity kymographs were able to firstly confirm our observations in *Figure 1E–G* of the differences in cell motion between 0% and 5% serum conditions. As shown in *Figure 3A*, all cell combinations grown in 0% serum exhibited a similar global motion pattern, minimal sheet-like motion and interface movement following gap closure. By contrast, in 5% serum the same three cell line combinations exhibited very different dynamics, amongst which we observed the formation of a stable equilibrium following contact only between EPC2 and CP-A cell sheets. However, subsequent computation of the mean sheet speed, as performed during routine PIV analysis, highlighted the analytical need for a computational framework to extract more descriptive motion measures. As shown in *Figure 3B*, aside from an indication of a global speed increase in 5% serum, speed was an insufficient metric to characterise the observed phenotypes, exhibiting high variance between replicates and failure to highlight the uniqueness of the EPC2:CP-A interaction. We thus used MOSES to construct more discriminative phenotypic measures to characterise the boundary formation of EPC2:CP-A.

Using MOSES, we constructed a motion saliency map for visualising global motion patterns (*Figure 3—figure supplement 2*) and designed four different additional measures as summarised in *Table 1* to quantitatively distinguish between different combinations of the three cell lines (EPC2, CP-A and OE33) used in our assay: i) boundary formation index, based on the degree that motion concentrates into a local spatial region (*Figure 3—figure supplements 2* and *3*); ii) mesh stability index, which measures the degree of movement between neighbouring cell groups locally within the sheet at the endpoint (*Figure 3—figure supplement 4*); iii) mesh order to measure the degree of collective motion of the whole sheet (*Figure 3—figure supplement 5, Video 5*) and iv) the maximum velocity cross-correlation, which measures the average correlation in motion between the two epithelial sheets from their movement history (*Figure 3—figure supplement 6*). The first three are based on the mesh; the fourth uses the individual superpixel tracks. These measures represent examples of statistics derivable using MOSES and are not limited to wound-healing type assays. For technical details, see Materials and methods. Cells grown in 0% serum were used as a computational negative control to set appropriate cut-offs for detecting boundary formation in our computational analysis (see Materials and methods for details).

Computing the proposed indices for all videos, the boundary formation index was ranked on a continuous scale for all 5% serum videos (*Figure 3—figure supplement 7*) by MOSES. The boundary formation index was highest (median 0.74) for EPC2:CP-A grown in 5% serum (n = 30/77) (*Figure 3C*, *Figure 3—figure supplement 8B*), whilst EPC2:EPC2 and EPC2:OE33 in 5% serum were below the boundary formation cut-off. In 5% serum, EPC2:CP-A had the highest mesh stability index

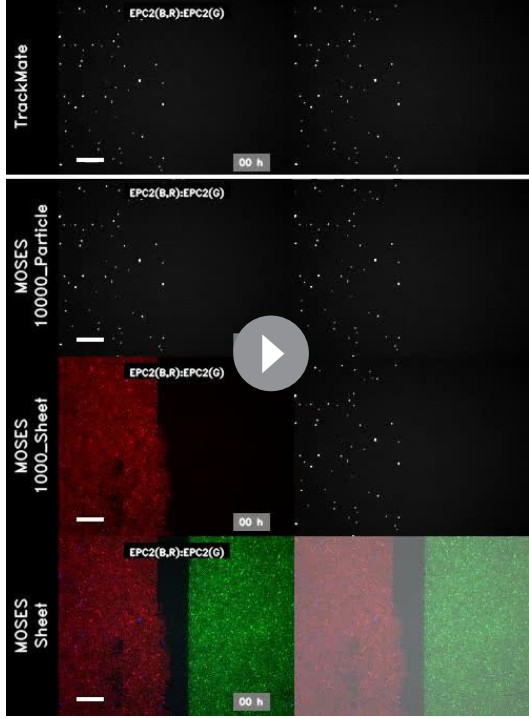

**Video 4.** Assessment of individual cell motion in confluent cell sheets using spiked-in fluorescent cell populations and MOSES. Blue (B) represents the spiked-in population of EPC2 cells. (R) and (G) refer to the red and green EPC2 epithelial sheets, respectively. Bar: 500 μm.
DOI: https://doi.org/10.7554/eLife.40162.014

(median 0.90), compared to EPC2:EPC2 (0.76) and EPC2:OE33 (0.25) (*Figure 3D*, *Figure 3—figure supplement 8C*). Together these results illustrate that only the squamous-columnar combination (EPC2:CP-A) formed a final stable boundary.

We next measured sheet-sheet interactions using the maximum velocity cross-correlation (VCC) before and after gap closure for the three cell line combinations. Gap closure was automatically determined (*Figure 3—figure supplement 9* , Materials and methods). For two initially diametrically opposed migrating sheets, a significant increase in VCC after gap closure compared to before gap closure is suggestive of increased sheet interaction. In 0% serum, no difference in velocity cross-correlation across all combinations was found before and after gap closure (*Figure 3E* ) - the two sheets do not move cohesively as a unit, as expected with minimal cell-cell contact. In serum, the difference for EPC2:CP-A (0.03 before and 0.20 after gap closure) was ~3-6 times larger than for EPC2:EPC2 (0.01 to 0.08) and EPC2:OE33 (0.02 to 0.05) (c.f. left and right violin plots in *Figure 3E* ), suggesting potential physical interaction. Interestingly, CP-A:OE33 (Barrett's:cancer, n = 6) also exhibited a substantial increase in VCC following gap closure (0.03 to 0.17) (*Figure 3—figure supplement 8D,E*) but no substantial increase was observed for CP-A: CP-A (0.01 to 0.06) (*Figure 3—figure supplement 8D,E*), confirming this observation was not simply CP-A cell line-specific. Both the velocity order parameter and our mesh order parameter (*Figure 3F,G*) indicate clear overall reduction in collective motion in cells grown in 0% serum compared to 5% serum.

Finally, we used MOSES with image segmentation and convolutional neural network (CNN) cell counting to assess morphological variations of the interface among all cell line combinations to check for 'invasive fingers' or boundary breakdowns that often exist in normal and tumour cell-formed boundaries (*Figure 3—figure supplement 10A,B*). Interestingly, we found non-invading boundaries in all cell line combinations except CP-A:OE33, where we observed clear infiltration with 'finger-like' protrusions of CP-A into the OE33 cell sheet grown in 5% serum (*Figure 3—figure supplement 10B*). No such 'finger-like' protrusions or significant intermixing of cells (*Figure 3—figure supplement 10D,F–H*) were detected in EPC2:CP-A under the same conditions. Altogether, our characterisation suggests that, among all tested cell-type combinations, we detect an 'interacting' final stable boundary uniquely in the squamous-columnar EPC2:CP-A combination grown in 5% serum (*Video 6*).

## MOSES can measure subtle phenotype changes induced by external stimuli

The ability to assess subtle phenotype changes with a minimal number of replicates is critical for high-content screening applications. We thus tested the ability of MOSES to detect changes in EPC2:CP-A boundary formation using an external perturbation. The epidermal growth factor receptor (EGFR) is frequently mutated in EAC, sometimes overexpressed in BE and is activated by bile acid reflux, the main observed cause of BE clinically (*Dixon et al., 2001*; *Souza, 2010*). In our experiments we thus used epidermal growth factor (EGF), the ligand of EGFR, as a biologically relevant external perturbation. Increasing amounts of EGF (0 ng/ml to 20 ng/ml) were added to the culture

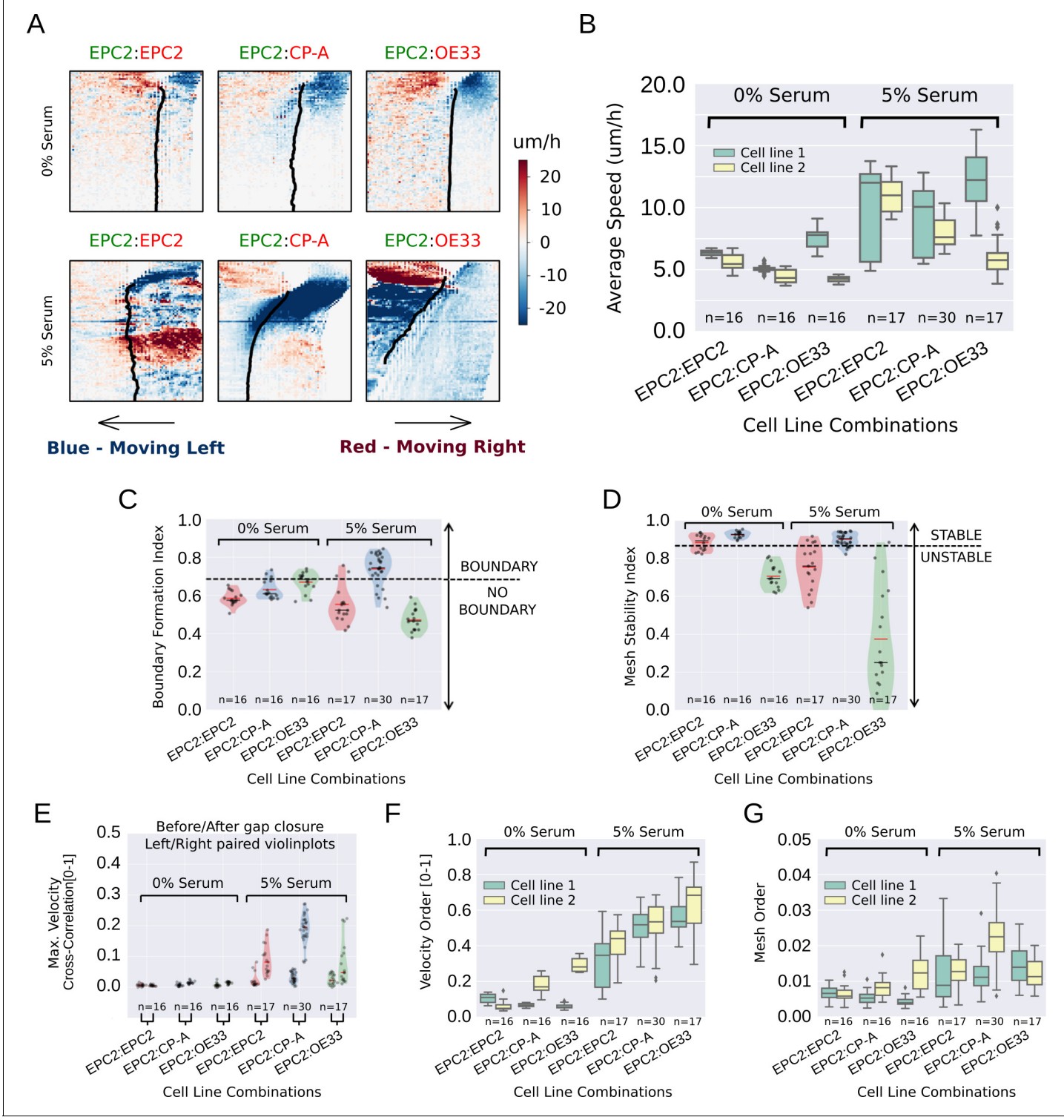

**Figure 3.** MOSES quantifies boundary formation and epithelial sheet interaction dynamics. (A) Projected 'x'-direction velocity kymograph of the dense optical flow for each cell line combination at the two serum concentrations used in the assay. The speed and direction of movement are indicated by the intensity and colour respectively (blue, moving left; red, moving right). (B) Average speed of each video for each cell combination, coloured green and yellow for the first and second cell line in the named combinations respectively. (C,D) Violin plots of boundary formation index (C) and mesh stability (D) for each video (black dot) for different cell combinations in 0% and 5% serum. Dashed line is the threshold, one standard deviation above the pooled mean value of all cell line combinations in 0% serum. Red solid line in violins = mean, black solid line in violins = median. (E) Maximum (Max.) velocity cross-correlation between the two sheets, before and after gap closure, left and right violins respectively for each cell line combination. *Figure 3 continued on next page*

*Figure 3 continued*

Shaded region of all violins is the probability density of the data whose width is proportional to the number of videos at this value. (F) Boxplots showing median and interquartile range (IQR) of velocity order parameter and (G) mesh order for each cell line combination coloured green and yellow for the first and second cell line in the named combination, respectively. Whiskers show data within 1.5 x IQR of upper and lower quartiles.

DOI: https://doi.org/10.7554/eLife.40162.015

The following figure supplements are available for figure 3:

**Figure supplement 1.** Heterogeneity in motion dynamics and quality of image acquisition.

DOI: https://doi.org/10.7554/eLife.40162.016

**Figure supplement 2.** The motion saliency map and boundary formation index for analysing motion sources.

DOI: https://doi.org/10.7554/eLife.40162.017

**Figure supplement 3.** MOSES mesh and boundary formation index captures multiple boundary formation.

DOI: https://doi.org/10.7554/eLife.40162.018

**Figure supplement 4.** The MOSES mesh stability index captures the stability of the local topology.

DOI: https://doi.org/10.7554/eLife.40162.019

**Figure supplement 5.** The mesh strain vector and collective motion.

DOI: https://doi.org/10.7554/eLife.40162.020

**Figure supplement 6.** Velocity cross-correlation (VCC) for measuring the motion coordination of two epithelial sheets.

DOI: https://doi.org/10.7554/eLife.40162.021

**Figure supplement 7.** Ranking of 5% serum videos according to boundary formation index.

DOI: https://doi.org/10.7554/eLife.40162.022

**Figure supplement 8.** Quantitative assessment of boundary formation and sheet-sheet interaction dynamics of all 5% serum videos.

DOI: https://doi.org/10.7554/eLife.40162.023

**Figure supplement 9.** Automatic determination of gap closure.

DOI: https://doi.org/10.7554/eLife.40162.024

**Figure supplement 10.** Cell infiltration, boundary shape and cell intermixing at the interface between two sheets.

DOI: https://doi.org/10.7554/eLife.40162.025

medium to assess incremental effects on cellular motion and boundary formation in the EPC2:CP-A combination. A total of 40 videos (each 144 h, one frame per h) were collected from three independent experiments, in a 24-well plate medium-throughput screen (*Supplementary file 2*). With increasing EGF, the boundary position was displaced a distance farther from the initial point of

**Table 1.** Summary of MOSES-derived measurements for discriminating boundary formation phenotype in this paper and their biological interpretation and application.

| Proposed measures | Biological interpretation | Biological application |
|---|---|---|
| Motion Saliency Map | 'Heatmap'-like image that highlights spatial regions that attract or repel local cellular motion over a defined time period. | Quantitative spatio-temporal readout of scratch or wound-healing assays. Highlights areas of salient motion activity spatially such as the migration of macrophages locally to inflicted wound sites. |
| Boundary formation index (value from 0 to 1) | Quantifies the concentration of movement within one region of space. Multiple 'hotspots' decrease this index. Uniform distributed movement (no attraction) scores 0. Concentration of movement in a single region for example a line or point scores 1. | Quantification of the spatial uniformity or 'spread' of motion activity. Highlights for example if all migrating cells move uniformly to close the wound in a wound-healing assay. |
| Mesh stability index (value from $-\infty$ to 1) | Measures the motion stability of local cell groups by measuring the change in relative spatial arrangement (topology) with respect to neighbouring cell groups at a given endpoint. Static epithelial sheets and epithelial sheets that move uniformly in a single direction preserve local cellular arrangement and scores 1. Motion that results in the local spatial rearrangement of cells such as neighbour exchange in embryos is unstable and scores < 1. | Assessing the final global movement stability of a collective such as an embryo or epithelial sheet based on the movement of the cells that compose it. |
| Mesh order (value > 0) | Measures the collectiveness of local cellular migration by the change in distance and direction of cells with respect to neighbouring cells. Hypothesizes that cells that are part of the same motion group exhibit the same 'mesh' force and move retaining the same local spatial arrangement. | Quantitative assessment of the extent of global sheet-like movement. |

DOI: https://doi.org/10.7554/eLife.40162.026

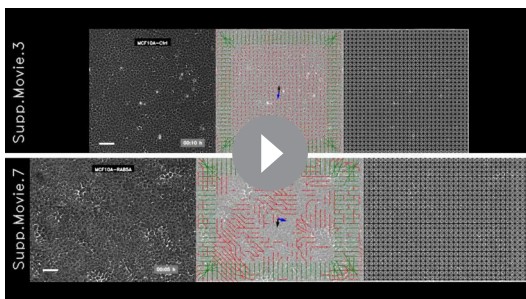

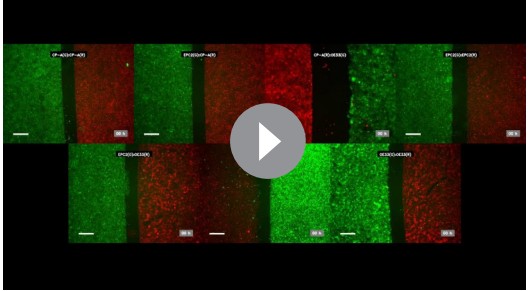

**Video 5.** Comparison of the temporal stability of velocity and mesh strain vectors for measuring collective motion. Individual velocity vectors are represented with red arrows, individual mesh strain vectors with green arrows. The large blue and black arrow are the global mean velocity and global mean mesh strain vector respectively. The corresponding derived MOSES mesh is shown on the right of the arrow plots in black. Bar: 100 μm.
DOI: https://doi.org/10.7554/eLife.40162.032

**Video 6.** Representative videos of the sheet migration of all tested cell combinations in 5% FBS. (R) and (G) denote red and green dyed cells respectively. Film duration 144 h. Bar: 500 μm.
DOI: https://doi.org/10.7554/eLife.40162.027

contact between the two cell populations, with slightly enhanced cell speed and decreased boundary coherence (*Figure 4A–D*). This is quantitatively reflected in the shape of the mean normalised strain curve (*Figure 4E*) which measures the average distance between neighbouring cell groups (Materials and methods): at 0 ng/ml EGF, the curve linearly increases before plateauing around 72 h; as EGF concentration increases, the curve becomes more linear and the plateau is lost above 5 ng/ml.

The boundary formation index decreased with increasing EGF (0.74 at 0 ng/ml to 0.46 at 20 ng/ml, comparable to EPC2:OE33 without EGF (0.46)), indicating loss of the boundary (i.e. index below the 0.69 cut-off) (*Figure 4F*). The mesh stability index decreased from 0.94 (stable, 0 ng/ml EGF) to 0.72 (unstable, 20 ng/ml) (*Figure 4G*), suggesting increased movement between neighbouring cells and the loss of interaction between the two cell populations since the maximum VCC difference before and after gap closure decreased from 0.16 (0 ng/ml EGF) to 0.04 (20 ng/ml EGF) (*Figure 4H*). The maximum VCC after gap closure was similar to that for EPC2:OE33 (0.02) (*Figure 4H*), but the mesh stability index remained higher (*Figure 4G*). Together these measures show that above 5 ng/ml EGF, the phenotype of EPC2:CP-A becomes similar to that between EPC2 and the EAC cell line, OE33 (*Video 7*). Cell counting and quantification of fluorescence decay suggest cell division is not the primary factor influencing the boundary at the cell density used in our experiments in 5% serum (*Figure 4—figure supplement 1*). Of note, titrating EGF did not rescue the effect of serum absence, with non-significant changes in the boundary formation index (0 ng/ml: 0.60 ± 0.07, 20 ng/ml: 0.65 ± 0.03) and maximum velocity cross-correlation, although the mesh stability index decreased, likely due to increased cell movement (*Figure 4—figure supplement 2* (n = 25)).

Using both the velocity order parameter and mesh order index to characterise collective motion, we found little change in collective motion upon EGF addition in 0% serum (*Figure 4—figure supplement 2J,K*), which might explain the lack of boundary formation. However, in 5% serum plus EGF, the two measures exhibited opposite results: raising EGF concentration increased velocity order but decreased mesh order, *Figure 4I,J*. We note however that the mesh order, by explicitly accounting for the motion of neighbouring cells, better reflects human observation of motion in videos (*Videos 5–7*). This highlights the pitfalls of only quantifying the individual alignment of velocity vectors computed from one video frame. Cell counting for 0% serum again suggested minimal influence of cell proliferation (*Figure 4—figure supplement 2L,M*). In summary, this example with EGF demonstrates that MOSES enables robust continuous-scale quantification of motion phenotype following systematic perturbation.

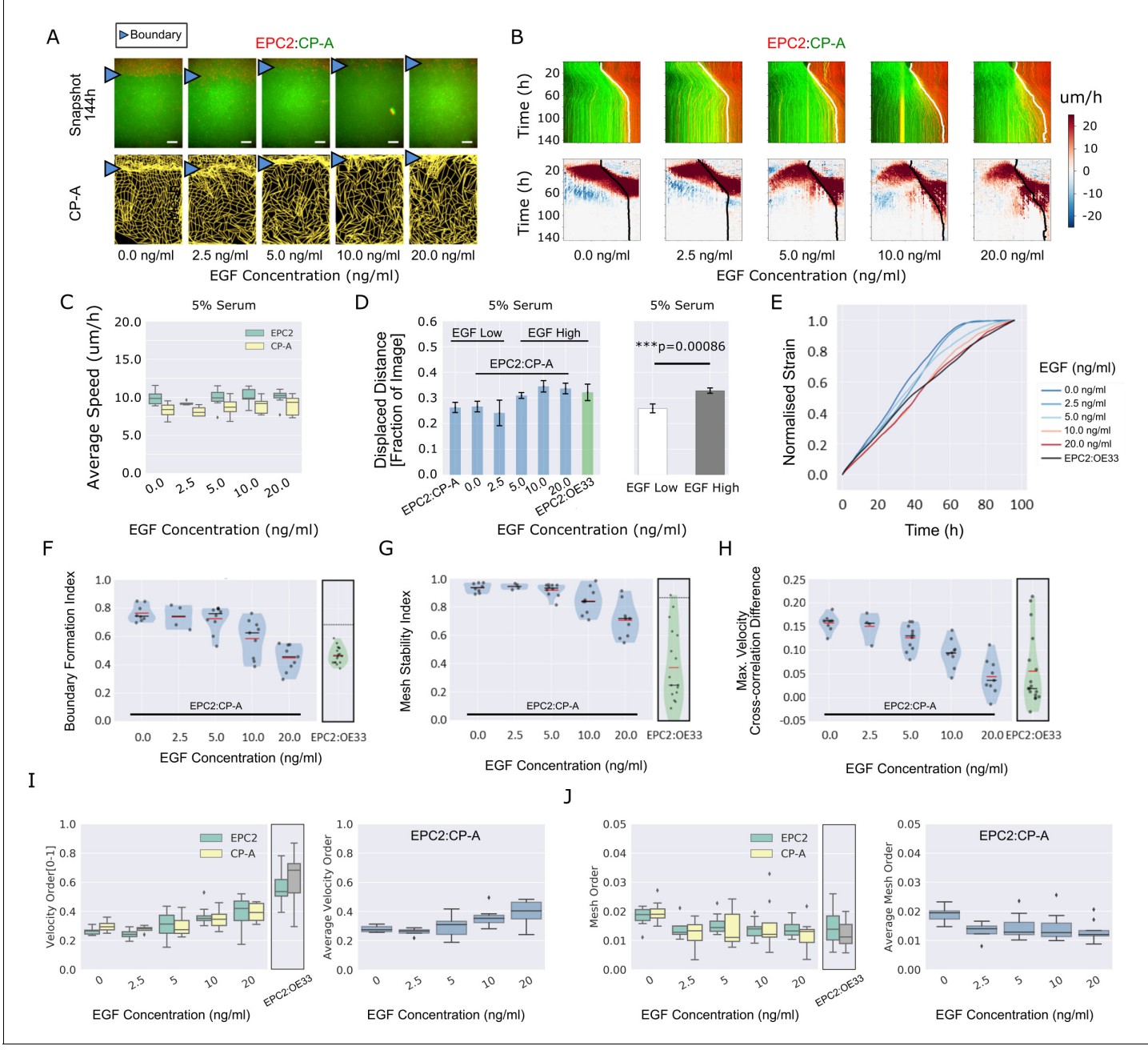

**Figure 4.** EGF titration at physiological levels disrupts boundary formation. (**A**) Destabilisation of the junction with EGF addition. All in 5% serum with snapshots of endpoint (144 h). Shown also is the green channel CP-A MOSES mesh. The closeness of the lines indicates impeded motion leading to a local aggregation of superpixels in the vicinity and is suggestive of a boundary. The less lattice-like the mesh, the less ordered the motion. Blue triangles mark the boundary position in the image and its corresponding inferred position in the CP-A mesh. All scale bars: 500 μm. (**B**) Top: maximum projected video kymograph. Bottom: x-direction velocity kymograph computed from optical flow for the representative videos in (**A**). (**C**) Grouped boxplot of the average speed for the different cell lines in the combination in 5% serum with increasing EGF concentration. (**D**) Mean displaced distance of the boundary normalised by image width following gap closure with increasing EGF concentration in 5% serum. Mean displaced distance of EPC2:CP-A and EPC2:OE33 cultured in 5% serum from **Figure 1G** are also plotted for comparison. T-test was used with * indicating p = < 0.05, ** p = < 0.01, *** p = < 0.001. Error bars are plotted for ± one standard deviation of the mean. (**E**) Mean normalised strain curves for EPC2:CP-A in 5% serum for each concentration of EGF. The mean curve for EPC2:OE33 videos in 5% serum without EGF in **Figure 3** is shown for comparison (black curve). (**F–H**) Violin plots of boundary formation index (**F**), mesh stability index (**G**) and maximum velocity cross-correlation (**H**) for each concentration of EGF and cells in 5% serum. Red solid line = mean, Black solid line = median. Dots are individual videos, total n = 40. Shaded region is the probability density of the data whose width is proportional to the number of videos at this value. Violins of respective measures for EPC2:OE33 in 5% serum without EGF with thresholds (horizontal black line) from **Figure 3** is shown for comparison. (**I,J**) Boxplots of velocity order (**I**) and mesh order

*Figure 4 continued on next page*

Figure 4 continued

(J) for individual cell lines (left) and pooled across the two cell lines in the combination (right). Values for EPC2:OE33 in 5% serum without EGF and threshold from **Figure 3** are shown for comparison.
DOI: https://doi.org/10.7554/eLife.40162.028

The following figure supplements are available for figure 4:

**Figure supplement 1.** Migration-independent cell counting to assess cell proliferation upon EGF addition to EPC2:CP-A in 5% serum.
DOI: https://doi.org/10.7554/eLife.40162.029

**Figure supplement 2.** EGF addition to EPC2:CP-A in 0% serum does not induce boundary formation.
DOI: https://doi.org/10.7554/eLife.40162.030

## MOSES generates motion signatures and 2D motion maps for unbiased characterisation of cellular motion phenotypes

High-content imaging screens are often explorative, with the aim of screening for unknown differences in complex cellular motions from a large number of videos in an unbiased manner (**Zaritsky et al., 2017**). In general it is therefore not known *a priori* the behaviour of the imaged cells. MOSES addresses this need for an unbiased phenotyping approach by enabling the systematic generation of unique 'motion signatures' for individual videos in a manner similar to the relatively automatic generation of geometric features for cell shape quantification in high-content image screens (**Boutros et al., 2015**; **Bray et al., 2016**). Below we demonstrate that unsupervised machine learning techniques requiring no manual user annotation can be applied to MOSES generated signatures to visualise all videos onto a 2D motion phenotype map. This advanced feature of MOSES enables easy visual assessment of motion phenotype and the generation of hypotheses without the need to individually interrogate each video.

The general process for the motion map generation is illustrated in **Figure 5A**. To position each video on a 2D map, we applied principal component analysis (PCA) to the normalised mesh strain curves of the 77 videos of all cell line combinations cultured in 5% serum conditions to learn the principal component vectors that define the *x-y* axis of the 2D map. The normalised mesh strain curve for each video was used here as an example 1D motion signature to summarise the entire video motion (see Materials and methods for constructing more descriptive signatures). The constructed PCA map of the 77 videos from cells cultured in 5% serum (**Figure 5B**) shows that this unbiased approach automatically clusters all videos corresponding to each cell line combination. Furthermore, the videos were ordered in a continuous manner, as shown by the increasingly linear shape of the mean normalised strain curve when looking left to right across the plot in **Figure 5B** from CP-A: CP-A to EPC2:OE33. This clustering was not achieved with root mean squared displacement (RMSD), the non-mesh version of the MOSES-normalised mesh strain curve (**Figure 5—figure supplement 1**). Moreover, it appears independent of the particular dimensionality reduction technique used (**Figure 5—figure supplement 2**), indicating that the signatures constructed using MOSES are intrinsically informative. Finally, the 1D MOSES-based motion signatures trained a machine learning classifier with no further processing to predict cell combination identity better than RMSD (**Figure 5—figure supplement 2**).

To demonstrate how such 2D motion phenotype maps can be used to compare videos, we next mapped the 48 videos from 0% serum cultures on the same axes as the videos from 5% serum cultures (**Figure 5C,D**). The videos from

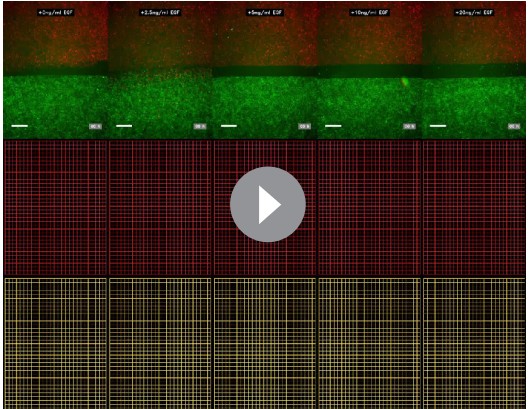

**Video 7.** Motion dynamics of EPC2(R):CP-A(G) under increasing EGF addition (0–20 ng/ml). MOSES can also measure decreased collective migration in the green CP-A sheet with increasing EGF, a phenomena difficult to assess by eye. MOSES meshes are shown, red for red-dyed (R) EPC2 cells and yellow for green-dyed (G) CP-A cells, respectively. Bar: 500 μm.
DOI: https://doi.org/10.7554/eLife.40162.031

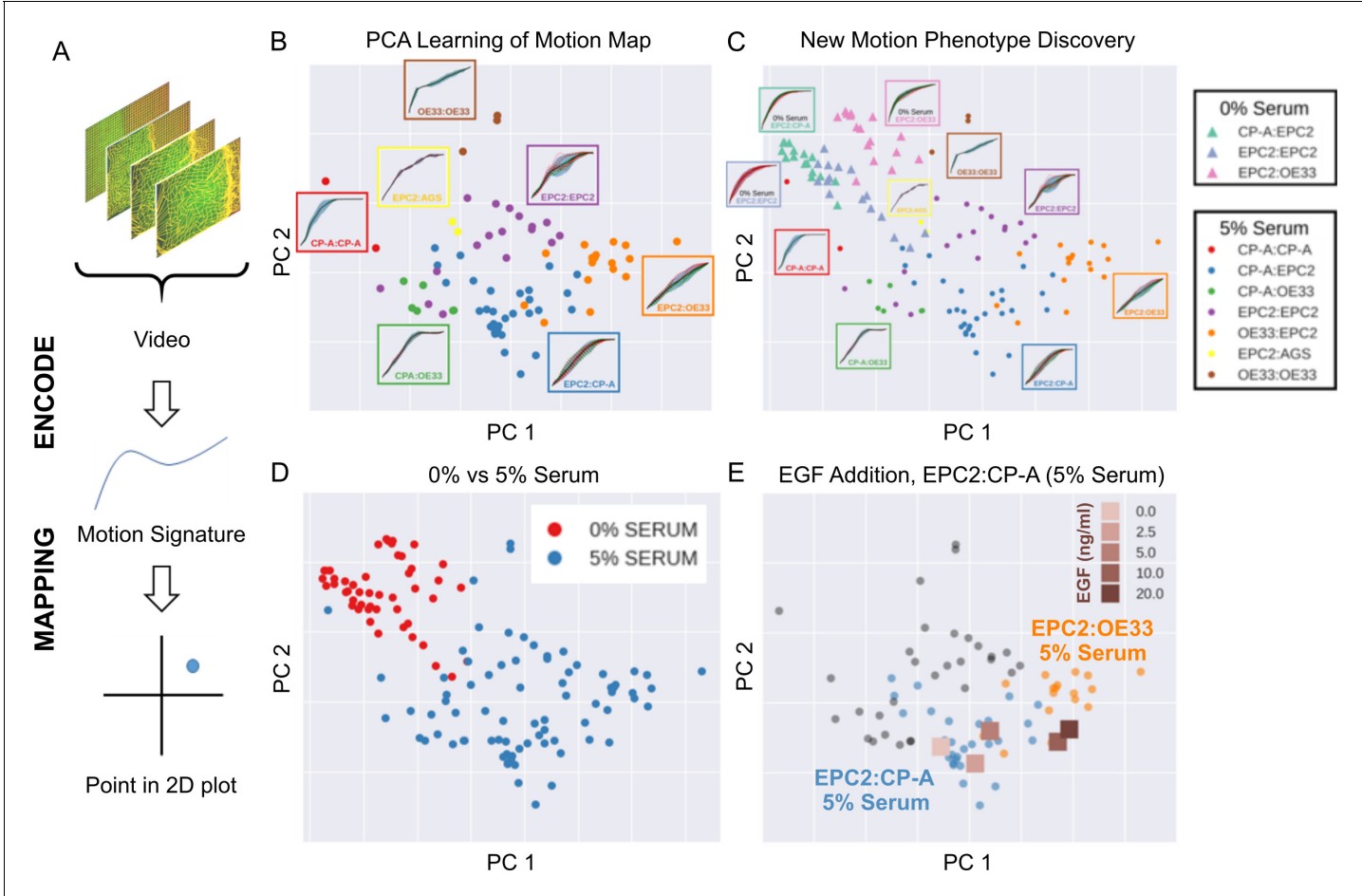

**Figure 5.** MOSES generates motion signatures to produce a 2D motion map for unbiased characterisation of cellular motion phenotypes. In all panels, each point represents a video (see legends for colour code). The position of each video on the 2D plot is based on the normalised mesh strain curves, analysed by PCA. (A) The mapping process for a single video. (B) The 5% serum videos (n = 77) were used to set the PCA that maps a strain curve to a point in the 2D motion map. (C) The 0% serum videos (n = 48) were plotted onto the same map defined by the 5% serum videos using the learnt PCA. In (B) and (C), the mean mesh strain curves for each cell combination are shown in the insets. Light blue region marks the two standard deviations with respect to the mean curve (solid black line). (D) Same map as in (C) with points coloured according to 0% or 5% serum. (E) The normalised mean strain curves for 0–20 ng/ml EGF addition to EPC2:CP-A from *Figure 4E* plotted onto the same map defined by the 5% serum videos.

DOI: https://doi.org/10.7554/eLife.40162.033

The following figure supplements are available for figure 5:

**Figure supplement 1.** Comparison of MOSES-normalised strain curves vs RMSD curves as motion signatures for motion map generation from 5% serum videos.

DOI: https://doi.org/10.7554/eLife.40162.034

**Figure supplement 2.** Comparison of motion map learning using different dimensional reduction techniques with MOSES strain curves and RMSD curves.

DOI: https://doi.org/10.7554/eLife.40162.035

0% serum mapped to a different area of the 2D plot, whilst preserving the continuous ordering of the previous videos. Therefore, without having watched the videos, it is easy to predict that the cells have markedly different motion dynamics in 0% serum compared to 5% serum. Furthermore, since the points for the 5% serum videos cover a larger area of the 2D plot than the 0% serum videos, one can predict more diversity of motion in 5% serum (*Figure 5D*).

The motion map can also capture subtle changes in dynamic behaviour. This is demonstrated by mapping the mean video motion for each concentration of EGF from 0 to 20 ng/ml (represented by the respective mean normalised strain curves for each concentration (one per concentration from a total n = 40 videos, *Figure 4E*) onto the same axis as the 5% serum videos in the absence of EGF

(square points in *Figure 5E*). With increasing EGF, the squamous-columnar EPC2:CP-A motion dynamics become increasingly similar to squamous-cancer EPC2:OE33 above 5 ng/ml, as evidenced by the square points moving from the area of blue circular EPC2:CP-A points into the area of orange circular EPC2:OE33 points. Thus our motion map is consistent with the result using the specific derived measures (above). These results illustrate the ability to detect biological and technical variability across independent experiments and that MOSES possesses the required features for an algorithm to be used in an unbiased manner in high-content screens with minimal prior knowledge.

## Comparison between MOSES and PIV

Finally, to further illustrate the full potential of MOSES and to demonstrate its application, we compared MOSES with the widely used PIV method on two published timelapse microscopy datasets of epithelial monolayers. In the original publication, *Malinverno et al. (2017)* used PIV to describe the induction of large-scale coordinated motility in MCF-10A RAB5A expressing cells compared to MCF-10A control cells. In the publication associated with the second dataset, *Rodríguez-Franco et al. (2017)* used PIV to show the detection of deformation waves that propagate away from the cell boundary between two epithelial monolayers. The MDCK cell monolayers expressed EphB2 and its ligand ephrinB1, respectively.

Reanalysing the datasets with MOSES, we found that motion fields inferred from optical flow by MOSES were similar to those from PIV, yielding both similar speed curves and velocity kymographs. However, MOSES exhibited greater sensitivity to salient motion events (indicated in *Figure 6A*). Compared to PIV velocity vectors, MOSES superpixel tracks are a more data-efficient (see Discussion) encoding of the spatio-temporal velocity distribution that naturally enhances and preserves the salient motion. Reconstructed velocity kymographs from the MOSES motion trajectories capture not only the pattern of the full velocity kymograph but further selectively enhanced the detection of the deformation wave signature formed at the interface between EphB2/ephrinB1 epithelial monolayers (as indicated in *Figure 6A*, right panel). Thus, all velocity-based statistics that can be derived from PIV, such as the velocity order parameter, are fully preserved in MOSES. Yet, MOSES offers additional advanced possibilities. Firstly, instantaneous velocity-based measures from single videos commonly derived from PIV are noisy. For example, the velocity order parameter variation for the slower MCF-10A control cells is non-smooth and highly variable between consecutive time points (*Figure 6B*). This leads to the misinterpretation that at certain time points, MCF-10A control cell motion is more collective than MCF-10A RAB5A expressing cells following induction. In contrast, the MOSES mesh order exploits long-time continuity and neighbourhood relations to robustly capture collective motion in a manner consistent with human observation (*Video 5*). Secondly, long-time MOSES tracks and superpixel mesh strain curves can unbiasedly cluster the global motion pattern into small spatio-temporal groups (*Figure 6C*, *Video 8*). This provides a systematic approach to the interrogation of motion sources (*Figure 6D*, *Video 9*). The computation of motion saliency maps from forward and backward tracked frames effectively highlight the spatial concentration of motion, and the boundary formation index attests to the efficacy of the MOSES-enabled measures across independent datasets. Finally, and uniquely, MOSES' meshes and tracks present a systematic framework for users to define extensive sets of custom measures to comprehensively characterise complex motion phenotypes, such as the investigated boundary formation behaviour between epithelial cells. By exploiting this, we successfully constructed feature descriptors to generate interpretable motion maps for our large video dataset. *Figure 6E* summarises the key points of comparison between MOSES and PIV.

## Discussion

We have shown that MOSES combines the advantages of existing PIV and single-cell tracking methods to provide a single systematic approach for the analysis of complex motion and interaction patterns. Its operating principle builds upon two established and highly successful approaches, long-time trajectories and graphs/meshes. In computer vision, spatio-temporal 'signatures' constructed using long-time trajectories have proven superior to 'signatures' derived from PIV-type motion fields for encoding complex spatio-temporal events such as human actions in very large datasets (*Wang et al., 2011*). Meshes/graphs are ubiquitously regarded as one of the best approaches to capture relationships between 'objects' across many disciplines, from Google search to protein-

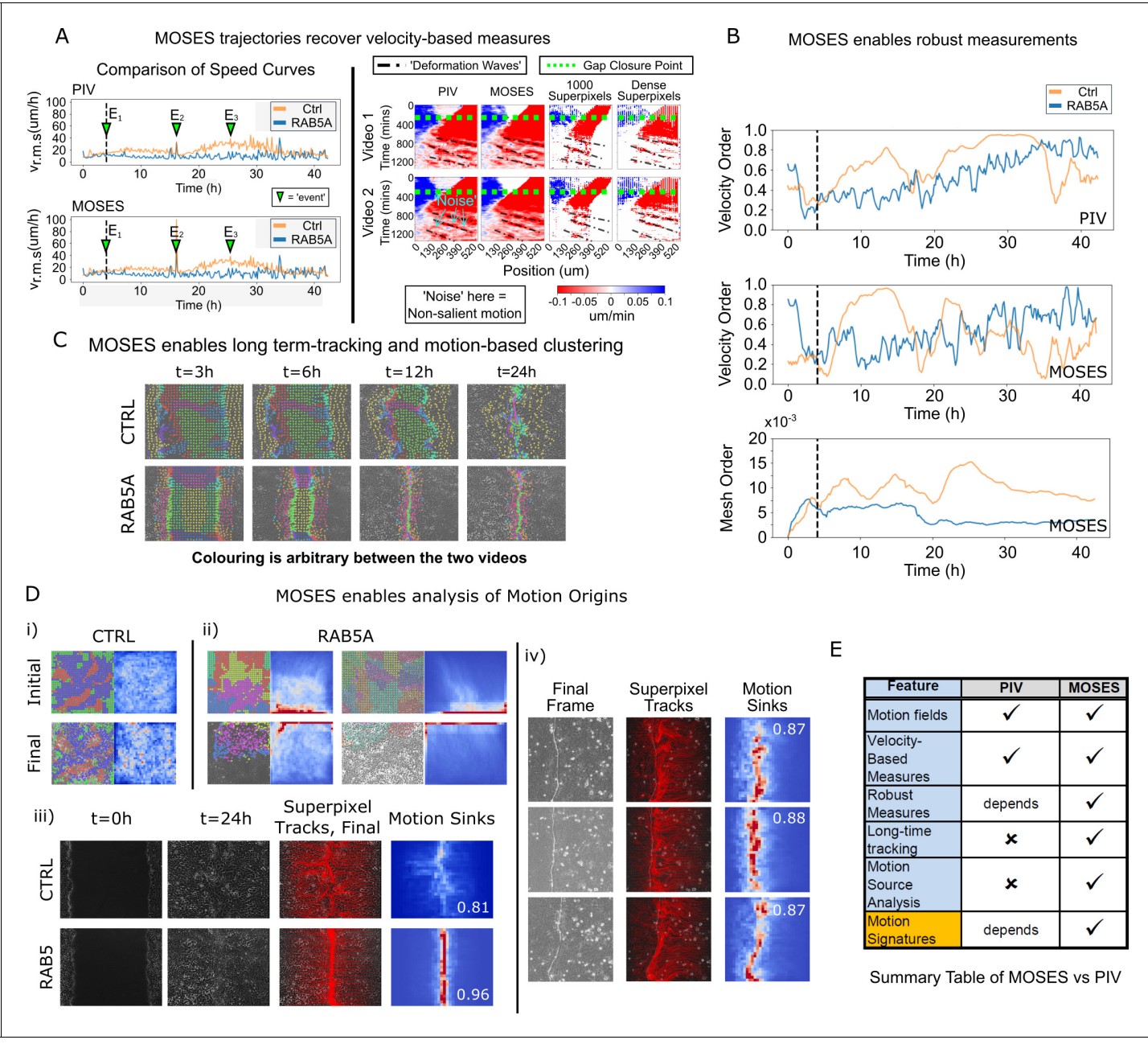

**Figure 6.** Comparison between MOSES and PIV. (**A**) Left: average speed curves of MCF-10A control (CTRL) and doxycycline inducible RAB5A-expressing (RAB5A) monolayer cell migration after doxycycline addition (Supp. Movie 3 of *Malinverno et al., 2017*) computed using PIV and MOSES (optical flow). Green triangles indicate notable events in the movie; E1 (4 h): addition of doxycycline, E2 (16 h): first bright 'flash' in movie followed by accelerated movement of RAB5A expressing cells, E3 (25 h): timepoint at which RAB5A cells moved fastest. Right: velocity kymographs of Videos 1 and 2 (c.f. Figures 3 and 4 respectively in *Rodríguez-Franco et al. (2017)*) computed from PIV and MOSES motion fields showing the presence of deformation waves (black dash-dot line) due to cell jamming following initial gap closure (green dashed line). The corresponding velocity kymographs reconstructed from a fixed number of 1000 MOSES superpixel tracks and a 'dense' number of superpixel tracks (starting with 1000 superpixels) is shown for comparison. The speed and direction of movement are indicated by the intensity and colour, respectively (blue, moving right; red, moving left). (**B**) Top and middle: velocity order parameter curves as defined in *Malinverno et al. (2017)* for the MCF-10A control and RAB5A expressing cell lines was computed for the same movies as (**A**) using PIV and MOSES. Bottom: corresponding MOSES mesh order curve. Black vertical dashed line mark the addition of doxycycline. (**C**) MOSES superpixel tracks computed for wound-healing assay of MCF-10A and RAB5A cells (Supp. Video 19 of *Malinverno et al., 2017*) were automatically clustered into distinct groupings and coloured uniquely according to their mesh strain curve using GMM with BIC model selection (Materials and methods, *Video 8* of this paper). (**D**) (i,ii) Automatic clustered superpixels and associated motion saliency maps (backward tracking to identify 'initial' motion, forward tracking to identify 'final' motion) for different videos of MCF10A-control (CTRL) and RAB5A monolayer migration in *Malinverno et al. (2017)*, (see also *Video 9*). (iii) Left to right: snapshots of initial and final frames, MOSES superpixel tracks

*Figure 6 continued on next page*

*Figure 6 continued*

(1000 superpixels) overlaid on final frames, associated 'final' motion saliency map and boundary formation index for CTRL and RAB5A cells for same videos in (C). (iv) Snapshot of final frame, overlaid MOSES superpixel tracks (1000 superpixels) and associated 'final' motion saliency map and boundary formation index for the boundary formation between EphB2 and ephrinB1 expressing MDCK monolayers (*Rodríguez-Franco et al., 2017*). (E) Summary of the comparison between MOSES and PIV. Long-time refers to the tracking of movement beyond one timepoint.

DOI: https://doi.org/10.7554/eLife.40162.036

protein interaction networks (*Szklarczyk et al., 2015*), flocking analyses (*Ballerini et al., 2008*; *Zhou et al., 2013*; *Shishika et al., 2014*), to detection of cell jamming in biological physics (*Lačević et al., 2003*; *Park et al., 2015*). MOSES uniquely brings together these disparate uses of long-time trajectories and meshes into one general analysis framework. As a result, MOSES satisfies the four criteria (robust, sensitive, automatic, and unbiased) necessary for characterising and establishing new phenotypes from live cell imaging. The analysis of datasets that include variable quality videos and experiments with a small number of replicates demonstrates the potential of the proposed computational framework.

Importantly, MOSES is progress towards overcoming the individual limitations of single-cell tracking and PIV-type velocity methods. Single-cell tracks are notoriously problematic over long times; the track of a single cell may be lost or broken into many separate tracks. MOSES superpixel tracks avoids this and recovers the global motion patterns (c.f. motion saliency maps, derived measures and motion signatures). Side-by-side comparison of MOSES and the standard PIV method using published datasets demonstrates that MOSES not only enables all the measurements of PIV, but by further exploiting long-time tracks and neighbourhood relationships, delivers greater physical and biological insights. Complex salient spatio-temporal motion patterns and events such as boundary formation, deformation waves due to cell jamming between two cell populations and cell death can all be quantitatively captured by MOSES. Critically, the ability of MOSES to perform long-time tracking (up to 6 days demonstrated in this study) enabled spatial localisation of the cell populations involved in a particular motion phenotype.

MOSES does not require complex user settings to facilitate reproducibility in analyses because it does not aim to threshold or cluster out the moving objects or phenotypes during analysis, which would introduce intermediate processing errors. Rather its philosophy is to facilitate systematic generation of many motion-related measurements based on trajectory and mesh statistics sufficient for applying machine learning methods for data-driven object segmentation, video classification and phenotype detection in large video collections (e.g. *Figure 5*, motion map) with minimum prior information. The main parameter the user specifies is the number of initial superpixels, which determines the spatial resolution of analysis. No complicated fitting of complex models and no special hardware such as GPUs are required. MOSES is

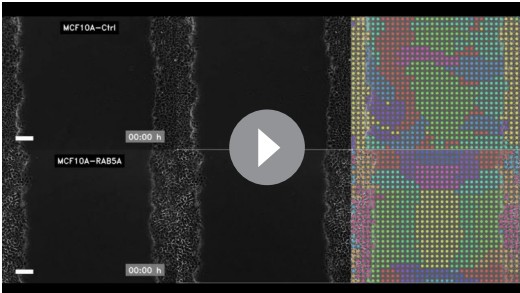

**Video 8.** Long-time superpixel track extraction and unbiased track clustering using the mesh strain curve of each superpixel as the feature vector for Supp. Movie 19 of *Malinverno et al. (2017)*. Each cluster is highlighted with a unique colour. The colours are arbitrary. There is no colour matching between individual videos. Bar: 100 μm.

DOI: https://doi.org/10.7554/eLife.40162.037

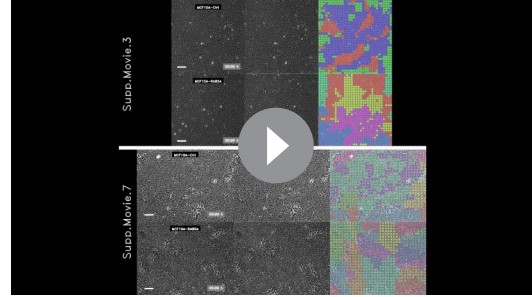

**Video 9.** Long-time superpixel track extraction and unbiased track clustering using the mesh strain curve for Supp. Movie 3 and 7 of *Malinverno et al. (2017)*. Each cluster is highlighted with a unique colour. The colours are arbitrary. There is no colour matching between individual videos. Bar: 100 μm.

DOI: https://doi.org/10.7554/eLife.40162.038

modular and its components can be readily adapted to suit specific applications, for example non-square superpixel shapes to better capture cells that undergo large shape changes. Analysis of 96 videos with 1344 × 1024 pixel resolution and 145 frames by tracking 1000 superpixels takes under 4 h on an unoptimized code implementation running on a single PC (3.2 GHz, 16 GB RAM). Results are stored efficiently (~1–2 MB per video) compared to ~0.1–1 GB per video, depending on the sub-sampling used to save the full spatio-temporal PIV/optical flow motion fields. Altogether our study illustrates the potential of MOSES as a powerful and systematic computational framework. It is particularly useful for unbiased explorative high-content screening with an aim to discover fundamental principles of cellular motion dynamics in biology and to identify factors or drugs that alter cellular motion dynamics in disease aetiology and treatment.

# Materials and methods

## Key resources table

| Reagent type (species) or resource | Designation | Source or reference | Identifiers | Additional information |
|---|---|---|---|---|
| Cell line (*H. sapiens*) | EPC2 | https://www.med.upenn.edu/molecular/documents/EPCcellprotocol032008.pdf | | Prof. Hiroshi Nakagawa (University of Pennsylvania) |
| Cell line (*H. sapiens*) | CP-A | CP-A (KR-42421) (ATCC ) | ATCC:CRL-4027; RRID:CVCL_C451 | |
| Cell line (*H. sapiens*) | OE33 | ECACC | ECACC:96070808; RRID: CVCL_0471 | |
| Cell line (*H. sapiens*) | AGS | AGS (ATCC CRL-1739) | ATCC:CRL-1739; RRID:CVCL_0139 | |
| Chemical compound, drug | KSFM | Gibco/Thermo Fisher | Cat#:17005042 | |
| Chemical compound, drug | RPMI 1640 medium | Gibco/Thermo Fisher | Cat#:21875–034 | |
| Chemical compound, drug | Human recombinant EGF | Gibco/Thermo Fisher | Cat#:PHG0313 | |
| Chemical compound, drug | Celltracker Orange (CMRA) | Life Technologies/ Thermo Fisher | Cat#:C34551 | |
| Chemical compound, drug | Celltracker Green (CMFDA) | Life Technologies/ Thermo Fisher | Cat#:C7025 | |
| Chemical compound, drug | Celltracker DeepRed | Life Technologies/ Thermo Fisher | Cat#:C34565 | |
| Chemical compound, drug | Image-iTTM FX Signal Enhancer | Thermo Fisher | Cat#:I36933 | |
| Chemical compound, drug | Antibody diluent, background reducing | Agilent Dako | Cat#:S3022 | |
| Chemical compound, drug | Antibody dilutent | Agilent Dako | Cat#:S0809 | |
| Chemical compound, drug | Fluoromount-G | SouthernBiotech | Cat#:0100–01 | |
| Chemical compound, drug | DAPI (1 mg/ml) | Thermo Fisher | Cat#:62248 | 1:1000 dilution |
| Antibody | Goat polyclonal anti-mouse Alexa Fluor 488 | Thermo Fisher | Cat#:A-11001; RRID:AB_2534069 | 1:400 dilution |
| Antibody | Phalloidin Alexa Fluor 647 | Thermo Fisher | Cat#:A22287; RRID:AB_2620155 | 1:400 dilution |
| Antibody | Mouse monoclonal anti E-cadherin | Becton, Dickinson U.K Ltd. | Cat#:610181; RRID:AB_397580 | 1:400 dilution |

*Continued on next page*

*Continued*

| Reagent type (species) or resource | Designation | Source or reference | Identifiers | Additional information |
|---|---|---|---|---|
| Other | 25 culture-inserts 2-Well for self-insertion | Ibidi | Cat#:80209 | |
| Software | Fiji ImageJ | https://imagej.net/Fiji | RRID:SCR_002285 | TrackMate Plugin |
| Software /Algorithm | Motion Sensing Superpixels | This paper | RRID:SCR_016839 | https://github.com/fyz11/MOSES |

## Cell lines and tissue culture

EPC2 (from Prof. Hiroshi Nakagawa, University of Pennsylvania, Perelman School of Medicine, Department of Gastroenterology, USA) and CP-A (ATCC) cells were grown in full KSFM (Thermo Fisher), AGS (ATCC) and OE33 (ECACC) in full RPMI with 10% FBS. Both were supplemented with glutamine and Penicillin streptomycin at 37°C and 5% $CO_2$ until 80% confluent. To passage EPC2 and CP-A, cells were resuspended after trypsinization for 5 min with PBS supplemented with soybean trypsin inhibitor (0.25 g/L, Sigma) to prevent cell death prior to resuspension in KSFM. To store, cells were resuspended at a concentration of $10^6$ cells/ml with 90% FBS +10% DMSO freezing media following centrifugation and stored at −80°C before passing to liquid nitrogen storage. All cell lines were tested monthly for mycoplasma infection using MycoAlertTM PLUS Mycoplasma Detection Kit (Catalog #: LT07-705 from Lonza) at the Ludwig Cancer Institute, Oxford, UK and have not shown evidence of *Mycoplasma*. Cell lines have been authenticated by Eurofins.

## Fluorescent labelling

Cells were labelled using Celltracker Green CMFDA and Celltracker Orange CMRA dyes (Life Technologies) according to protocol. Two different concentrations 2.5 µM and 10 µM were used. The lower concentration still permits tracking but has fewer toxicity concerns.

## Immunofluorescence staining

Samples were washed twice with PBS and fixed with 4% PFA for 15 min. The samples were then washed twice and permeabilised with 0.1% Triton-X for 15 min. Samples were blocked with image-iT Fx signal enhancer for 1 h before incubation with E-cadherin (610181 Becton Dickinson U.K. Limited) overnight at a 1:400 dilution in Agilent Dako Antibody Diluent with Background Reducing Components (S3022, Agilent Dako) at 4°C. Following 3 washes of 10 min in PBS, samples were incubated for 2 h in the dark with Alexa 488 goat anti-mouse (A11001, Thermo Fisher) secondary antibody and Alexa Fluor 647 Phalloidin (A22287, ThermoFisher), both at a 1:400 dilution and DAPI (1 mg/mL) at 1:1000 dilution (62248, Thermo Fisher) in Agilent Dako Antibody dilutent (S0809, Agilent Dako). Finally, samples were washed three times with PBS and mounted using Fluoromount-G (0100–01, SouthernBiotech).

## Temporary divider Co-culture assay

In 70 µL of culture media, 70,000 labelled cells were seeded into each side of a cell culture insert (Ibidi) in one well of a 24-well plate. After 12 h, inserts were removed and the well washed three times with PBS to remove non-attached cells before adding the desired media (KSFM (0% serum in the text) or 1:1 mixture of KSFM:RPMI + 5% FBS (5% serum in the text)) for filming. For the perturbations, the effector, for example EGF, was also added to the media at the stated concentrations.

## Spike-in experiments

A third population of cells dyed with 2.5 µM of CellTracker DeepRed (Life Technologies) was diluted 1:200 into one of the other two populations dyed with either Celltracker Green CMFDA (Life Technologies) or Celltracker Orange CMRA (Life Technologies) of the same cell line. The mix was then added to one side of the insert (Ibidi) as with the two cell population experiments described in the main text.

## Image acquisition

The different conditions were filmed on a Nikon microscope for 96 or 144 h at a frequency of 1 image per hour. 2x and 4x objectives were used. The microscope filter wavelengths used to visualise the red and green dyes were 546 nm and 488 nm, respectively.

## Automated cell counting with convolutional neural networks

The convolutional neural network (CNN) density counting approach of *Xie et al., 2016* was used to automatically count cells in confocal DAPI-stained nuclei images or fluorescent and phase contrast video frames. Once trained, given an input image the CNN model outputs a dot-like image with the property that the sum of all pixel intensities in the output dot-like image equals the number of cells within the image (*Figure 1—figure supplement 1A*). We describe the training method for DAPI images only. Other image modalities were trained in a similar manner details of which are given below under 'Migration independent cell counting in videos'. To generate the training data for DAPI cell counting, 200 image patches (size 256 pixels x 256 pixels) were first randomly extracted from the acquired DAPI images of resolution 4096 pixels x 4096 pixels (*Figure 1—figure supplement 1A*). For each extracted 256 × 256 image patch, the stained nuclei centroids were manually marked using a 'dot'. Then for each 256 × 256 image patch, 50 randomly sampled 64 pixel x 64 pixel patches were extracted to yield a total training set of 10,000 image patches (*Figure 1—figure supplement 1A*). The CNN training settings used for a 70:30 train-test split were 200 epochs, batch size 100, RMSprop (lr = 0.001, rho = 0.9, epsilon = 1e-08, decay = 0.0) with a mean absolute error (MAE) loss. The final test accuracy was reported as the mean absolute deviation (MAD) between manually counted and predicted cell counts on the result of applying the learnt CNN to the larger 256 × 256 manually labelled image patches (*Figure 1—figure supplement 1B,C*). To count specific cell types stained with different coloured dyes from confocal images, epithelial sheets were segmented from their respective channels by applying k-means clustering on the RGB image pixel intensity values with k = 3 (retaining the two clusters of highest intensity) after downsampling the full-size images (4096 × 4096 pixels) by a factor of 4 (to 1024 × 1024 pixels). Small objects (<200 pixels) were removed, holes were filled and the largest connected component kept before upsampling the binary mask to its original resolution (4096 × 4096). The respective final binary mask was used to mask and count cells specific to each colour channel (*Figure 1—figure supplement 1C*). Counting of specific cell types in timelapse video frames, which are more variable in quality, is similar but uses the more optimized image segmentation protocol detailed below to segment individual epithelial sheets.

## Migration-independent cell counting in videos

To count the migrating cells in the videos, two different approaches were used. The first and default approach used in this paper is the automatic CNN counting described above trained on manual annotations of the fluorescent video frames (*Figure 4—figure supplement 1A,B*). To bypass the issue of moving areas, we first produced a binary mask based on image segmentation as described below to identify the respective 'red' and 'green' cell sheet areas to quantify. Using the mask, equal sized image patches of 64 × 64 were randomly sampled. These were then fed into the trained CNN to produce cell count estimates for each image patch. The average cell density over 100 random such image patches were taken as an estimate of cell density for the entire frame. This operation was repeated for each channel separately and for every time point to yield a temporal cell count profile. The proliferation rate was subsequently estimated as the average absolute cell density change in successive frames normalised by the mean cell density (computed over the desired time frame) (*Figure 4—figure supplement 1C*). To check results were unaffected by the imaging modality, CNN based counting was also applied to the corresponding phase contrast images where the fluorescence channels were used to identify the individual cell types (*Figure 4—figure supplement 1D*). The second approach exploits the change in dye fluorescence as cells proliferate (the cell dye intensity decreases as they divide) (*Figure 4—figure supplement 1E–H*). For our videos, the image intensity exhibits too discrete a transition between time points (*Figure 4—figure supplement 1F*) for accurate extraction of the fluorescence decay. Instead, given that all videos of EGF addition to EPC2:CP-A were of the same temporal length (144 h), we found that the modal image intensity and modal normalised frequency of the intensity histogram peak at every time point served as an alternative quantification of fluorescence decay that yields the same conclusions (data not shown) but could

be robustly fitted with a linear best-fit line (*Figure 4—figure supplement 1I*). The faster the fluorescence decay, the higher was the cell proliferation and the higher the absolute value of the fitted linear gradient that we used as a proxy proliferation coefficient (*Figure 4—figure supplement 1J,K*).

## Image segmentation of epithelial sheets in timelapse video

Red and green channel images were anisotropically filtered (*Perona et al., 1994*) to enhance image edges and suppress stochastic image noise. The entropy image was then computed to enhance the epithelial sheet. Otsu thresholding was subsequently applied to obtain red and green binary masks of the sheet. Finally 'holes' in the resultant masks were filled using a line rastering approach.

## Computing the distance displaced of the boundary following gap closure

A custom image edge finding script using Sobel filters to detect edges on downsampled fluorescent images was used to find the leading sheet edge (in terms of an average $x$-coordinate) of both cell populations for each time frame (the averaged $y$-coordinate of the boundary did not change significantly over the filmed duration). The gap closure point was determined by the intersection of the two $x$-coordinate curves. The displaced distance of the boundary following gap closure was computed as the absolute distance between the final frame averaged $x$-position and the gap-closure frame averaged $x$-position.

## MOSES framework

MOSES was developed using the Python Anaconda 2.7 distribution, in particular it uses Numpy, the Scipy-stack and OpenCV libraries. It comprises separately a cell tracking and data analysis component.

## Motion extraction

Regular superpixels were generated by applying the SLIC (*Achanta et al., 2012*) algorithm in scikit-image on a blank image, the same size as the video frame. 1000 superpixels were used throughout in this paper. Motion fields for updating the superpixel centroid position over time were computed with OpenCV Farnebäck optical flow (*Farnebäck, 2003*) using default parameters. For ease of implementation, displacement vectors were rounded to the nearest integer. Superpixels passing out of the frame progressively lose pixels and retain their last motion position for the tracking duration. For simplicity, lost area is not recovered. PIV was computed using the Python openpiv package, using the closest equivalent parameters to MOSES; window size of 16 pixels, overlap of 8 pixels and search area of 32 pixels.

## Motion feature-based sheet segmentation and superpixel assignment

Step 1: Given the tracks of one colour for example red, identify all superpixels that initially move by thresholding on the cumulatively moved distance within the first few frames, (here we used 2 frames, equivalent to 2 h). Step 2: Form the superpixel neighbourhood graph by connecting together all identified moving superpixels from step 1 to any other identified moving superpixel within a specified radial distance cutoff (1.2 x average superpixel width here) using their initial $(x,y)$ centroid positions. The largest connected graph component is then found to approximate the covered area of the epithelial sheet at frame 0. Step 3: In some videos, image artefacts such as autofluorescence or the presence of isolated cells contributes superpixel tracks that are biologically irrelevant and affects quantification of the migrating sheet dynamics. Tracks associated with these noise sources must be removed. The need for such removal is automatically evaluated through a user-set cutoff based on prior knowledge of the expected maximum fraction of the field-of-view covered by any one of the red or green populations at time $t = 0$ (e.g. for a 50:50 plating of red and green cells, a conservative cut-off fraction of 0.70 is used here). Assuming that no red and green superpixel can jointly occupy the same $(x,y)$ position in frame 0, joint filtering based on the degree of movement is applied to clean segmentation errors from previous steps. Step 4 (not required if running MOSES using dense tracking): The kept superpixels after steps 1-3, are then iteratively dynamically propagated to identify 'activated' superpixels (those that lie in the joint area occupied by the kept superpixels) frame-by-frame. Step 5 (potentially optional): In case the dynamic superpixel propagation identifies

superpixels that do not move much over the entire video and to ensure the temporal continuity of superpixel positions, steps 1-2 are repeated. Step 6: To ensure the same number of superpixel tracks across all videos for statistical comparison, finally we assign constant tracks for all unused or inactivated superpixel tracks where for all frames their $(x,y)$ positions are fixed to their initial centroid position. The procedure described is illustrated more concisely in *Figure 2—figure supplement 1A*.

## Dynamic mesh generation

To generate meshes, each superpixel is connected to its nearest 'neighbours'. The notion of 'neighbour' is mathematically defined by the user (see below). For the MOSES mesh used here for visualisation and stability analysis, we defined 'neighbours' using a pre-set distance cut-off based on the distance between individual superpixel centroids at the start of tracking. The mesh strain curve thus measures the relative distortion between connected superpixels with respect to their initial mesh geometry over time. For computation of the boundary formation index, a different mesh was used where neighbours were independently determined frame-by-frame by a pre-set distance threshold. The specified threshold in both meshes are given in Euclidean distance as a multiplicative factor of the average superpixel width used. A factor of 1.2 for the mesh strain and 5.0 was used for the boundary formation index throughout.

## MOSES dynamic meshes – mining contextual relationships of spatio-temporal tracks using geometry and graph theory

Numerous ways exist to connect a collection of $(x,y)$ points to form a graph or mesh. Depending on the mesh formed, different aspects of the movement can be enhanced and probed in interesting ways. In this paper, the presence of collective motion biologically motivates the mesh concept. More generally, meshes are 'abstract' constructs to assess relationships. In terms of motion analysis, in this paper, we recognise that different spatially located points may be correlated particularly if they are spatially close. In view of this, we first describe and explain the rationale of the two meshes constructed in the main text for visualising and quantifying epithelial sheet dynamics before describing possible extensions and how these may be more useful for specific experiments. The first and the primary mesh used throughout is what we term the MOSES mesh. It is constructed by joining each superpixel track with all superpixel tracks whose initial $(x,y)$ centroids are within a user-specified constant Euclidean distance cut-off. Implicitly this assumes that under collective sheet migration, initially spatially close superpixels continue to remain spatially close. Violation of this assumption leads to large stretching or compression of the MOSES mesh, which can be used to derive a measure of motion collectiveness (see mesh order below). The second mesh is used to generate the motion saliency maps for localising boundary formation, a dynamic state that varies frame to frame. Each superpixel track is again connected to all superpixel tracks whose $(x,y)$ centroids are within a user-specified Euclidean distance cut-off. However, contrary to the first mesh, which uses only the centroid positions in frame 0, here the distances are determined using the current position of all superpixels in the current frame. Thus the 'neighbours' continuously change frame-to-frame. We refer to this mesh as the radius neighbour graph and the motion saliency map is the result of counting the number of neighbours for each superpixel (i.e. the node degree). The number of surrounding neighbours increases in spatial areas where motion concentrates e.g. there exists a local chemoattractant or a physical impedance to cell movement such as a boundary. Thus boundaries are natural attraction motion centres; leading superpixels at the boundary cannot advance whilst those behind continue to move towards the boundary. This leads to an overall accumulation of superpixels at the boundary (c.f. motion saliency maps in *Figure 3—figure supplement 2*). In both meshes, tuning the distance cut-off threshold tunes the length-scale of the spatial interaction one wishes to analyse. For boundary formation, a phenomenon that spans the entire height of the image, one should choose a relatively large distance cut-off such as 5x the average superpixel width compared to, for example, identifying the localisation of a macrophage to a cell apoptosis site. In the latter, the attraction site is more point-like and a radius cut-off 1x the average superpixel width may be more accurate. Finally, whilst both of the presented meshes in our paper only utilise physical distance to define neighbours this is by no means the only possibility. In some applications such as flocking, the topological distance (i.e. the number of points away) not the physical distance may be more relevant (*Ballerini et al., 2008*). In this situation, it is more common to construct the local kNN (k-nearest

neighbour) graph, designating the closest k superpixels as neighbours. The kNN is also frequently used when the magnitude of the 'interaction' between points is not known *a priori* as a way to propagate local information and mine patterns in data c.f. t-SNE for dimensionality reduction, spectral clustering and similarity network fusion (*Wang et al., 2014*) for combining multimodal datasets. Finally, superpixels may be joined not only spatially but also temporally to enforce consistent temporal neighbour relations (*Chang et al., 2011*) as well as according to more 'semantic' notions of similarity for example similar image appearance, similar instantaneous velocities (*Chang et al., 2011*). In short, by formalising motion analysis under the framework of dynamical meshes that connects together 'neighbouring' superpixels, we can analyse complex motion not just in terms of instantaneous speed and orientation but can additionally leverage powerful established tools developed in the fields of computational graph theory, network theory, topology etc. to effectively quantify and mine increasingly complex motion phenotypes in high-content screens.

## Mean squared displacement (MSD)

As a measure of cellular motions, MSD was computed as a function of time interval, $\Delta t$ as in (*Park et al., 2015*).

$$\mathrm{MSD}(\Delta t) = \left\langle |\mathbf{r}_i(t + \Delta t) - \mathbf{r}_i(t)|^2 \right\rangle$$

where $\mathbf{r}_i(t)$ is the position of the superpixel $i$ at time $t$ and $\langle \cdot \rangle$ is the average over all time $t$ and all superpixels. For small $\Delta t$, the MSD increases as a power law $\Delta t^\alpha$, where the exponent $\alpha$ is determined empirically by fitting. For unity exponent ($\alpha = 1$), the movement is uncorrelated random Brownian motion and cellular motion is diffusive. When $\alpha > 1$, cellular motions are super-diffusive, and when $\alpha = 2$, motions are 'ballistic'.

## Root Mean Squared Displacement (RMSD)

As a summary of the whole video motion and a measure of movement, the root mean squared displacement was computed as a function of time relative to the initial time $t_0$, $\mathrm{RMSD}(t) = \sqrt{\left\langle |\mathbf{r}_i(t) - \mathbf{r}_i(t_0)|^2 \right\rangle}$ where $\mathbf{r}_i(t)$ is the position of the superpixel $i$ at time $t$ and $\langle \cdot \rangle$ is the average over all time $t$ and all superpixels. For multi-channel videos, the average RMSD was used to describe video motion. Unless otherwise stated in the text, the normalised RMSD (here division by maximum value within the common time window of comparison) was plotted to permit comparison across different conditions. To compare across videos of different duration, in *Figure 1—figure supplement 4B* we instead compute the RMSD divided by its value at 96 h, the maximum timepoint shared by both our 96 h and 144 h videos.

## Normalised spatial correlation

For each superpixel $i$ we define its neighbourhood, $N_i$ as all superpixels $j$ which lie within a specified distance, $r$. Given the time-dependent velocity function $V \equiv V(t) = \mathbf{r}(t + 1) - \mathbf{r}(t)$ where $\mathbf{r}(t)$ is the track (all $(x, y)$ positions) up to time $t$, the normalised spatial correlation of a video with a total of $n$ superpixel tracks is defined as

$$\frac{1}{n}\sum_{i=1}^{n} E_{\forall j \in N_i}\left[\frac{cov(V_i V_j)}{\sigma_{V_i}\sigma_{V_j}}\right]$$

where $E[\cdot]$ is the mean function averaging over the superpixel neighbourhood, $cov(\cdot)$ is the covariance function and $\sigma_{V_i}$ is the standard deviation of $V_i(t)$. Computing spatial correlation as a function of $r$ for our videos yields an exponential decay which can be fitted to an equation of the form $y = ae^{-x/b}$ from which the initial correlation $a$ and characteristic correlation distance $b$ can be determined for plotting, (*Figure 1—figure supplement 5*). In our plots, the distance $r$ is in terms of normalised units (i.e. the number of average superpixel widths away).

## Manual vs MOSES comparison on cell tracking challenge datasets

MOSES does not explicitly handle individual cells leaving and entering the field of view or cell divisions during filming. For fair comparison of motion capture ability with manually annotated tracks,

the tracks were only compared for cells present in the initial frame as depicted with coloured masks (*Figure 2B*). Single-cell tracks were generated from MOSES (1000 superpixels) by identifying the superpixel track that has moved the greatest distance over the video duration amongst all superpixel tracks whose initial ($x,y$) position lies within the area of the individual cell of interest as marked out by manual annotation at $t = 0$ (*Video 3*).

## TrackMate single-cell tracking

The Fiji TrackMate plugin (*Tinevez et al., 2017*) was run on the third blue image channel containing only the sparse population of DeepRed dyed cells with the following settings: estimated blob diameter, 10 pixel (default); threshold, 2.5; linking max distance, 50; gap-closing max distance, 50; and gap-closing max frame gap, 100. All other parameters were left at their default values.

## TrackMate vs MOSES comparison

As with the cell tracking challenge dataset for fair comparison of motion capture ability with single cell trackers like TrackMate, tracks were only compared for cells present in the initial frame as detected by TrackMate. To generate single-cell tracks using MOSES (with 10,000 superpixels) for the sparse DeepRed dyed cells, the nearest four superpixel tracks to each cell centroid were averaged to produce a single track. Similarly, to generate single cell tracks using MOSES (with 1000 superpixels) from the Green CMFDA dyed or Orange CMRA dyed sheet, for each cell, the nearest four green/red superpixel tracks were found to compute a mean track. Track similarity was computed by evaluating the normalised velocity cross-correlation (value between 0 and 1 as defined below) between each MOSES track and its corresponding TrackMate track with the average normalised velocity cross-correlation over all tracks reported for each video (*Figure 2—figure supplement 2B*, denoted M. for matched tracks). To assess the statistical significance of the resultant value, the track similarity from random pairing of the tracks were computed and the average of 10 permutations were reported (labelled P. for permuted in *Figure 2—figure supplement 2B*). All three combinations of cell types (EPC2:EPC2, EPC2:CP-A and EPC2:OE33) and all red/green dye combinations were tested, a total of 23 videos (each 144 h acquired with one image per h). The frame size of each acquired video was 512 × 672 pixels. As such, the mean spiked-in cell diameter was 5 pixels, the average superpixel width was 6 pixels (10,000 superpixels) and 19 pixels (1000 superpixels). From *Figure 2—figure supplement 2B* and *Video 4*, MOSES achieves near perfect similarity compared to TrackMate. In some cases, the produced MOSES tracks are more desirable, guaranteeing a continuous track whereas TrackMate requires explicit linking of cell detections across frames and thus often tends to produce many 'broken' tracks when the same cell is unable to be detected across all time points.

## Motion saliency map

The motion saliency map illustrates in a heat map format spatial areas of motion sources and sinks, and was constructed using the MOSES dynamic meshes and superpixel tracks. It is inspired by Lagrangian fluid mechanics (*Shadden et al., 2005*; *Ali and Shah, 2007*). To compute this map for each frame, the radius neighbour graph was constructed (see above paragraph) using the spatial positions of superpixels in that frame and a blank image was populated at the ($x,y$) centroid position of each superpixel with the count of the number of surrounding neighbours according to the radius neighbour graph. This yielded an 'image' of size n_frames x n_rows x n_cols, where n_rows, n_cols are the video frame dimensions. We now have a multidimensional spatial heat map for each frame that captures the spatial-temporal motion saliency. To reveal long-time temporally persistent behaviour, we averaged the heatmap both spatially and temporally using the superpixel partition of the initial video frame as illustrated in *Figure 3—figure supplement 2*. By construction, this spatial map is general for studying any phenomenon where spatial localisation plays a role.

## Quantification cut-offs

We assume normally/t- distributed statistics for all measures. Boundary formation cut-off was set one standard deviation above the pooled mean of 0% serum samples. Mesh stability index was set one standard deviation below the pooled mean of 0% serum EPC2:EPC2, EPC2:CP-A samples.

## Boundary formation index

The boundary formation index (*Figure 3—figure supplement 2*) quantifies the extent to which motion concentrates into localised spatial regions in the motion saliency map as a signal-to-noise ratio with value from 0 to 1 suitable for global comparison across video datasets. For example a boundary concentrates motion along a 'line' whilst cell death may generate multiple spot-like concentrations (*Figure 3—figure supplement 3*). The higher the index, the more that motion is concentrated into a single spatial region. To compute the boundary formation index from the visual motion saliency image, the motion saliency image was segmented into 'high' and 'low' intensity using Otsu thresholding and the normalised signal-to-noise ratio was computed, defined by $\frac{\text{mean(high)}-\text{mean(low)}}{\text{mean(high)}}$

(*Figure 3—figure supplement 2*). The mean was used as the motion saliency map was computed from a sparse set of points given by the number of superpixels. Individual pixel statistics such as the maximum intensity are therefore noisy and not robust measures. The denominator was set to be the mean of the 'high' intensity region in order to give a numerical value bounded between 0–1 for standardised comparison. As this measure captures the 'peakiness' of the spatial distribution, it can also be used to quantify other localised spatial processes with adaptation for example cell death.

## Normalised mesh strain curves

For a superpixel, $i$ at time $t$ we define the mesh strain, $\varepsilon_i(t)$ of the local neighbourhood, $N_i$ with $n$ neighbours as the mean of the absolute difference in the distance between superpixel $i$ and superpixel $j$ in its neighbourhood at time $t$, $r_{ij}(t)$ and at the start at $t=0$, $r_{ij}(0)$ so that $\varepsilon_i(t) = \frac{1}{n}\Sigma_{j\in N_i}\left|r_{ij}(t) - r_{ij}(0)\right|$ where $|\cdot|$ is the absolute value or *L1*-norm. The time-dependent mesh strain for one mesh is the mean local neighbourhood mesh strain over all superpixels for each time frame. The result is a vector the same length as the number of frames in the video. For multi-channels, the average vector is used to describe video motion. The absolute value of the strain curve is susceptible to the image acquisition conditions and geometry whilst the motion information is primarily encoded by the shape of the resulting curve. To permit comparison across different conditions, the normalised strain curve (here division by maximum strain within the common time window of comparison, 0-96 h here) up to 96 h (the maximum timepoint shared between 96 h and 144 h videos in this paper) was used as a simple signature to describe the global video motion pattern when computing motion maps.

## Normalised mesh strain, L1-norm and robustness

Here we provide more details as to why we chose the *L1*-norm for computing the mesh strain. When computing the 'distance' between two vectors, $\boldsymbol{x}_1$, $\boldsymbol{x}_2$ both of length $n$ one can define the notion of distance in different ways. A popular family of distances or norms to use is the $L^p$-norm denoted $||\boldsymbol{x}_1 - \boldsymbol{x}_2||_p$ defined mathematically as follows:

$$||\boldsymbol{x}_1 - \boldsymbol{x}_2||_p = \left(\sum_{i=1}^{n}|x_1^i - x_2^i|^p\right)^{\frac{1}{p}} \text{ where } i = 1,\ldots,n \text{ is the } i\text{th element of the vector}$$

$|\cdot|$ denotes the absolute difference. Within this family the most popular is the *L1* $(p=1)$ and *L2* $(p=2)$ norms. The *L1*-norm is also called the mean absolute deviation and the *L2*-norm the Euclidean or mean squared distance. The quadratic function of *L2* amplifies large differences (> 1) and reduces the effect of smaller differences close to zero thus where differentiability is not a concern *L1* is the more robust choice. In our case where most of the distances are larger than 1 (as all superpixels are initially seeded at a distance > 1), we chose to use *L1* which is more resistant to the effect of extremal values as a result of errors in motion extraction.

## Mesh stability index

The mesh stability index attempts to quantify the global stationarity (*Figure 3—figure supplement 4*) across a moving collective such as an epithelial sheet by measuring the change in the average distance between each superpixel and its neighbours. Specifically, it measures the flatness of the normalised mesh strain curve as defined above. Note whilst a curve is flat only if its gradient is flat that is 0, it is more visually appealing to report increasing stability with increasing values therefore

we define the mesh stability index as 1 minus the end gradient. The end gradient is most stable when it has a value of 0 therefore this index is upper bounded by 1. To standardise the value of the gradient for comparison, we normalise also across time such that for any length video, time is from 0 to 1. To compute the end gradient stably without curve fitting procedures, we assume the end point is locally linear with respect to time and average the first-order differences over the last few frames. For 96 h videos, the period of stability given by the curve plateau is shorter, therefore the last 10 frames (10 h) were used for computing the gradient. For 144 h videos, the last 24 frames (24 h) were used.

## Mesh order

Inspired by the definition of the velocity order parameter, the mesh order is identically defined but uses the local resultant mesh strain vector instead of instantaneous velocity vectors for computation (*Figure 3—figure supplement 5*). In this paper, we used the MOSES mesh but any similar mesh construction is equally valid. Given a mesh, the mesh strain vector for a superpixel is the sum of the displacement vector of the superpixel relative to each neighbouring superpixel (*Figure 3—figure supplement 5*). The mesh order is computed for each frame accordingly for a video. The mean value over all frames was reported for statistical comparison.

## Normalised velocity cross-correlation

Cross-correlation measures the similarity between two signals taking into account time delay, (*Figure 3—figure supplement 6*). As 'signals', we use the time-dependent velocity $V(t)$ computed from the spatial displacement between the location $r(t+1)$ at time $t+1$ and location $r(t)$ at time $t$, $V(t) = r(t+1) - r(t)$ which is spatially location-independent. Velocity is a vector quantity with both $x$ and $y$ components. Letting $\cdot$ denote the scalar product, the normalised velocity cross-correlation (VCC) of the tracks of red superpixel $i$ and green superpixel $j$ at time $t$ and time lag $m$ such that the extremas are bound by [-1,1] is

$$\text{VCC}_{ij}(m,t) = \frac{1}{T} \sum_{m=-T}^{T} \widehat{V}_i(t+m) \cdot \widehat{V}_j(t)$$

where $\widehat{V}_i = \frac{V_i - \bar{V}_i}{\sigma_i}$, $\bar{V}_i$ and $\sigma_i$ is the mean and standard deviation of $V_i$ respectively. $T$ is the maximum time lag and is set to be the length of $V_i$. VCC can be either positive or negative. We report the maximum absolute value for all red-green pairings and the average over all pairings as evidence of interaction between two epithelial sheets. In the main text, this is computed with tracks before (up to - 5 frames) and after (from + 5 frames) the gap closure point. The offset of 5 frames either side is based on the accuracy to which we could determine the gap closure point automatically (see below).

## Automatic gap closure determination

The frame of gap closure when the two epithelial sheets contact was found by finding the video frame in which the average distance between the migrating fronts of the two epithelial layers was minimised. To compute this distance as a function of time (frame number), first each epithelial sheet was independently segmented based on their respective colour channel pixel intensity (*Figure 3—figure supplement 9A*). For each sheet, images were preprocessed using a median filter (square kernel, the same size as the average superpixel width) and segmented using two class k-means clustering (for 0% serum, three class k-means was used to include the weaker pixel intensity of leading cells). The resulting segmentation was post-processed by binary morphological operations (binary closing with disk kernel of 5 pixels, removal of small objects (< 5% total image size) and binary filling). Sheet boundary points were efficiently identified using a sweepline algorithm. The image was evenly divided into 100 horizontal strips or sweeps in the *y*-direction. For each sweep, one boundary point was identified by selecting either the right-most point (largest *x*-coordinate) if the sheet was moving right, or left-most point (smallest *x*-coordinate) if the sheet was moving left (*Figure 3—figure supplement 9B*). Doing this for both sheets, each of the boundary points was paired in the red/green sheet to the closest in the opposing colour by physical distance. The average distance between the migrating fronts of the two epithelial layers for a particular frame was then the average euclidean distance of all red-green boundary point pairs in the frame. Computing the average

distance between the two sheets frame-by-frame, there was a change in the rate of decrease as the gap closed (*Figure 3—figure supplement 9C*). To estimate the frame at which this transition occurs, asymmetric least means squares (*Eilers and Boelens, 2005*) was used to first fit the baseline (representing the contribution to the distance measurement due to image segmentation errors) and second a linear spline (smoothing factor 0.1*variance(curve)) was used to approximate the temporal average distance curve. The gap closure frame was then found as the first frame for which the fitted spline value falls below the fitted baseline + 2*std(fitted baseline), where std is the standard deviation operation. We validated the method using a total of n = 246 videos of different cell combinations in different media by comparing the inferred frame to the consensus (average frame) of two manual annotators. A strong Pearson correlation coefficient of r = 0.902 (*Figure 3—figure supplement 9D*) and an accuracy of 94% within ± 5 frames (*Figure 3—figure supplement 9E*) compared to an accuracy of 97% within ± 5 frames between two human annotators (*Figure 3—figure supplement 9F*) was found.

## Boundary detection

Boundaries were detected either i) by image segmentation (early timepoints) or ii) from the MOSES tracks (late time points). For image segmentation, red and green epithelial sheets were segmented as described above for timelapse video frames. The boundary binary mask was then found as the mathematical set intersection of red and green binary masks. To derive a boundary line, the non-zero image coordinates of the binary boundary mask were forced to form a line given by unique $(x,y)$ coordinates by returning the average $x$-coordinate (along image horizontal) for each unique $y$-coordinate (along image vertical). A piece-wise cubic spline was subsequently fitted to enable interpolation of the boundary line. To derive the boundary line from the MOSES superpixel tracks, first all non-moving tracks that is all tracks that do not move a total distance greater than a predefined threshold was removed. Then at each frame, $t$ we considered all superpixels that have moved since the previous frame $t-1$. We then attempted to match the red superpixels to green superpixels where we defined a match when the distance between two points is smaller than a predetermined cut-off. All red and green superpixel points that have at least one successful match were kept. Together these points form the boundary candidate points. As discussed, a boundary is a motion attractor thus we can robustly find the boundary points from the candidate points using an asymmetric least squares filter (*Eilers and Boelens, 2005*). We modified the original formulation of *Eilers and Boelens, 2005* to enforce density based filtering by using a cut-off based on the number of neighbours within a predefined distance. Finally, given the boundary points the boundary line was found as in the case of image segmentation.

## Cell infiltration

For a single video frame, the infiltration of the first colour cell type into the second colour cell type is defined as the fraction of first colour cells that lie on the side of the boundary line of the second coloured sheet. The boundary line was located from image segmentation as described above.

## Boundary shape

The boundary shape is defined as the total length of the boundary line ($L$) divided by the equivalent straight line length ($L_0$) illustrated as solid white and yellow lines respectively in *Figure 3—figure supplement 10C*.

## Intermixing coefficient

We computed two different measures to reflect the 'intermixing' behaviour of cells at the boundary, which we denoted either '(Image)' or '(MOSES)' in *Figure 3—figure supplement 10*. The image intermixing coefficient reflects the 'spread' of the wound after we perform image segmentation on the video frame. A heavily infiltrated and wavy boundary yields a larger binary mask and occupies a larger fraction of the image than a sharp, linear boundary. We capture this idea by defining the intermixing coefficient (image) as the area of the boundary mask relative to the total image area. This is however a static measure of intermixing. It does not account for the fact that the boundary is not static and can be highly dynamic such as in EPC2:EPC2 or how EPC2 cells appear to 'flee' OE33 cells with little coordination. We thus propose a second dynamic measure of intermixing coefficient based

on measuring the spread in the motion behaviour. This intermixing coefficient (MOSES) is the area of the binary mask after thresholding the motion saliency map relative to the total image area. Of note, the two intermixing coefficients are identical if the final boundary is stable and the motion at the boundary during boundary formation is coordinated such as in EPC2:CP-A (*Figure 3—figure supplement 10E*).

## Kymographs

Boundaries were detected using the MOSES tracks as described. The kymograph of one time slice shows the median values of the projected *x*-direction velocities of all pixels as a function of the *x*-position. For superpixel tracks, the *x*-axis was binned with two times the number of unique *x*-coordinates based on the initial superpixel centroids (in line with Nyquist sampling theorem). This procedure was repeated for each timepoint to build the full kymograph over time.

## Generating feature descriptors for motion map generation

A feature descriptor for an image/video is a 1D numerical vector designed to summarise the image/video content in a compact code or signature. No universally optimal method exists to generate such a descriptor for all applications. In the main text, for simplicity to avoid the introduction of new concepts we used the normalised strain curve used to compute the mesh stability index as an example video feature vector for PCA. As we showed, this was sufficient to distinguish the different migration behaviour of the investigated cell combinations. However, the normalised strain curve is a coarse 1D approximation of the local MOSES mesh stretching and only one characteristic of the complex mesh dynamics. More generally, one can exploit other mesh constructions as discussed above to derive a plethora of graph theoretic measures such as the algebraic connectivity, the Laplacian spectrum and centrality, or supplement mesh-based statistics with trajectory-based measures such as turning angle and speed for a more comprehensive unbiased description of the spatio-temporal motion. In circumstances where cellular appearance exhibits large temporal changes such as in the case of migrating cells with lamellipodium, motion features alone may not provide a sufficiently descriptive signal for quantifying phenotypic differences. Here one could additionally supplement motion signatures with appearance-based features such as image texture descriptors (e.g. LBP, Haralick, HoG (histogram of oriented gradients) and SIFT (and its variants)) and construct the mesh semantically with the augmented motion-appearance descriptor.

## Dimensionality reduction experiments

We used the Python scikit-learn implementation of PCA (principal components analysis), MDS (multi-dimensional scaling) and TSNE (t-distributed stochastic neighbour embedding) and applied them to the 'raw' normalised mesh strain curves (no pre-processing). PCA was applied with n_components = 2 without input whitening. MDS used default scikit-learn parameters with n_components = 2, random_state = 0. TSNE used n_components = 2, learning_rate = 100, random_state = 0, init = 'random'. A 97-20-2 fully connected neural network autoencoder was implemented and trained with Keras (Tensorflow backend) with mean squared error loss using the Adam optimizer (lr = 0.001, beta_1 = 0.5, beta_2 = 0.999, epsilon = 1e-08, decay = 0.0). 48/77 of the 5% serum videos were used for training and the remaining 29/77 for validation to check for model overfitting. Tanh activations were used throughout. To maximise gradient propagation in the linear range of the 'tanh' activation function, we subtract 0.5 from the input normalised strain curves (values between 0 and 1) as a pre-processing step for the neural network.

## Automatic clustering of superpixel tracks

To automatically cluster superpixel tracks, we first computed for each superpixel its local mesh strain curve with respect to its neighbours. This yields a matrix $N$ rows by $T$ columns for $N$ superpixel tracks and a total of $T$ timepoints. Gaussian mixture model (GMM) was then used to generate clusters using BIC (Bayesian Information Criterion) to select the optimal number of clusters (*Fraley, 1998*).

## Software availability

The MOSES code is available open-source under a Ludwig non-commercial and academic license at GitHub, https://github.com/fyz11/MOSES.git (*Zhou, 2019*; copy archived at https://github.com/

elifesciences-publications/MOSES) where it is maintained and updated. Example data for testing can be downloaded from Google Drive, https://drive.google.com/open?id=0BwFVL6r9ww5BaTh6-NExLR1JMUXM. The full video dataset can be found under DOIs, https://dx.doi.org/10.17632/j8yrmntc7x.1, https://dx.doi.org/10.17632/vrhtsdhprr.1 for videos with and without EGF addition, respectively.

## Acknowledgements

We thank Prof. Hiroshi Nakagawa for generous donation of EPC2 cells and Profs. Roberto Cerbino and Xavier Trepat for making available the raw videos in their publications. We thank Dr. Mary Muers, Dr. Françoise Howe, Profs. Sebastian Nijman, Francis Szele, Colin Goding and Shankar Srinivas for critical reading of the manuscript; Mark Shipman for technical assistance with timelapse microscopy. This work is mainly funded by the Ludwig Institute for Cancer Research (LICR) with additional support from a CRUK grant to XL (C9720/A18513). FYZ is mainly funded through the EPSRC Life Sciences Interface Doctoral Training Centre EP/F500394/1, CRP and XL are funded by LICR, RPO is supported by LICR, the Oxford Health Services Research Committee and Oxford University Clinical Academic Graduate School supported by the National Institute for Health Research (NIHR) Biomedical Research Centre based at the Oxford University Hospitals Trust, Oxford, MJW is supported by CRUK (C5255/A19498, through an Oxford Cancer Research Centre Clinical Research Training Fellowship), and JR is funded by LICR and the EPSRC SeeBiByte Programme Grant (EP/M013774/1). The views expressed are those of the authors and not necessarily those of the NHS, the NIHR or the Department of Health.

## Additional information

### Competing interests

Felix Y Zhou, Carlos Ruiz-Puig and Jens Rittscher, Xin Lu: A patent is pending for MOSES (UK application no. GB1716893.1, International application no. PCT/GB2018/052935)). The code is available open-source and free for academic and non-profit users under a Ludwig academic and non-profit license. The other authors declare that no competing interests exist.

### Funding

| Funder | Grant reference number | Author |
|---|---|---|
| Ludwig Institute for Cancer Research | | Felix Y Zhou<br>Carlos Ruiz-Puig<br>Richard P Owen<br>Jens Rittscher<br>Xin Lu |
| Engineering and Physical Sciences Research Council | EP/F500394/1 | Felix Y Zhou |
| Oxford Health Services Research Committee | | Richard P Owen |
| Oxford University Clinical Academic Graduate School | | Richard P Owen |
| Cancer Research UK | C5255/A19498 | Michael J White |
| Engineering and Physical Sciences Research Council | EP/M013774/1 | Jens Rittscher |
| Cancer Research UK | C9720/A18513 | Xin Lu |

The funders had no role in study design, data collection and interpretation, or the decision to submit the work for publication.

## Author contributions
Felix Y Zhou, Conceptualization, Software, Formal analysis, Validation, Investigation, Visualization, Methodology, Writing—original draft, Project administration, Writing—review and editing, Conceived and implemented all of MOSES and analysed all data; Carlos Ruiz-Puig, Conceptualization, Resources, Data curation, Validation, Investigation, Methodology, Writing—review and editing, Performed all live cell imaging videos. Established the 24-well plate based co-culture system.; Richard P Owen, Michael J White, Resources, Methodology, Writing—review and editing, Aided initial prototyping of MOSES with mixed co-culture timelapse videos; Jens Rittscher, Conceptualization, Supervision, Methodology, Project administration, Writing—review and editing, Provided computational supervision; Xin Lu, Conceptualization, Supervision, Funding acquisition, Methodology, Writing—original draft, Project administration, Writing—review and editing, Provided overall project supervision

## Author ORCIDs
Felix Y Zhou http://orcid.org/0000-0003-4463-1165
Carlos Ruiz-Puig http://orcid.org/0000-0002-9214-3804
Xin Lu https://orcid.org/0000-0002-6587-1152

## Decision letter and Author response
Decision letter https://doi.org/10.7554/eLife.40162.047
Author response https://doi.org/10.7554/eLife.40162.048

# Additional files

## Supplementary files
• Supplementary file 1. Table summary of video datasets and experiments analysed for 0% and 5% serum.
DOI: https://doi.org/10.7554/eLife.40162.039

• Supplementary file 2. Table summary of video datasets and experiments analysed for EGF addition.
DOI: https://doi.org/10.7554/eLife.40162.040

• Transparent reporting form
DOI: https://doi.org/10.7554/eLife.40162.041

## Data availability
All data generated or analysed during this study are included in the manuscript and supporting files. Source code files for MOSES are available and maintained on GitHub (copy archived at https://github.com/elifesciences-publications/MOSES). Analysed videos are made available online with unique DOIs ( http://dx.doi.org/10.17632/j8yrmntc7x.1, http://dx.doi.org/10.17632/vrhtsdhprr.1) through Mendeley datasets.

The following datasets were generated:

| Author(s) | Year | Dataset title | Dataset URL | Database and Identifier |
|---|---|---|---|---|
| Zhou FY | 2018 | Normal 2 Cell Population Migration Dataset | http://dx.doi.org/10.17632/j8yrmntc7x.1 | Mendeley Data, 10.17632/j8yrmntc7x.1 |
| Zhou FY, Puig CR | 2018 | EGF Addition to EPC2:CP-A | http://dx.doi.org/10.17632/vrhtsdhprr.1 | Mendeley Data, 10.17632/vrhtsdhprr.1 |

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
