## [Decision Letter]

[Editors’ note: a previous version of this study was rejected after peer review, but the authors submitted for reconsideration. The first decision letter after peer review is shown below.]

Thank you for submitting your work entitled "Motion Sensing Superpixels (MOSES) is a systematic framework to quantify and discover cellular motion phenotypes" for consideration by *eLife*. Your article has been reviewed by three peer reviewers, and the evaluation has been overseen by a Reviewing Editor and a Senior Editor. The following individuals involved in review of your submission have agreed to reveal their identity: Roberto Cerbino (Reviewer #3).

As you will note, the three expert reviewers find some merit in the MOSES method but raise many issues that preclude publication. As we feel the additional work needed to address the issues raised by the reviewers would take more than two months to complete, we are returning your submission to you now in case you wish to submit elsewhere for speedy publication. Further, the number and complexity of these issues make the editorial team reluctant to encourage a resubmission unless a very substantial amount of additional work was performed. In this case, *eLife* would be willing to look at a revised paper. Please note that it would be treated as a new submission with no guarantees of acceptance. The two most critical factors from an editorial perspective are:

i) Demonstrating the utility of the method on a diverse range of imaging inputs (reviewer #3 point #5 – there is also a reference to some suggested data) and;

ii) Increased biological measurements, including analysis of proliferation and/or cell density. Given that this study is focused on a new technique, the final decision on this work will be highly dependent on convincingly demonstrating the broad utility of the methodology to the cell migration and tissue morphogenesis community. In addition, it will be important to address the technical issues raised, including out of plane movement or loss of cells, further analysis of the 'boundary', and improving the readability of the study.

Our decision has been reached after consultation between the reviewers.

Reviewer #1:

In the manuscript "Motion Sensing Superpixels (MOSES): A systematic framework to quantify and discover cellular motion phenotypes", Zhou et al. present a new method for measuring cell motility in epithelial sheets and apply this to boundary formation in Barrett's Esophagus.

The approach presented fills the technical gap between PIV/Optical flow measurements and single-cell tracking and provides a method to characterise, and through PCA, discover, motion phenotypes within 2D epithelia. This is certainly a technically interesting concept that is relatively simple to implement and provides insight into motion phenotypes within a tissue. This method will be broadly applicable to many problems in characterising cellular motion. In general I find the manuscript well written and comprehensive in its description of the technique, but not particularly easy to read.

One of the difficulties is that it is unclear to me whether the authors intend to reveal new insight into the fundamental phenomena or simply provide a framework for automating the analysis for e.g. drug discovery. I would argue that cell state (sub cellular protein levels, cell cycle, cell death/extrusion, as examined in Schmitz et al., 2010, Pau et al., 2013, Held et al., 2010) are equally important and are not addressed by the current method.

In the present manuscript, only the cell motion is considered and yet cell proliferation/density is likely to be very important also. Can the authors provide cell density measurements (a proxy for cell proliferation) as a function of time, perhaps using the CNN counting approach, to give some estimate of motion versus proliferation?

This would be particularly interesting in light of the experiments involving titration of EGF. These are very interesting and provide some insight into how collective dynamics/cell adhesion are important in the tissue mechanics in general.

Another consideration is that of axial/upward motion at the interface between the two epithelial monolayers as they meet. Presumably the MOSES approach would register very little motion in a region where cells are being extruded upward (as in Figure 1B). How do the authors envisage addressing this?

Overall, I think that this is a technically interesting manuscript, which provides a set of tools for measuring and characterising cell motion phenotypes from high-throughput time-lapse imaging. This is likely to be a broadly useful tool for the community and addresses a long-standing problem of characterising mesoscale behaviour with close to single-cell accuracy. In my opinion, a revised manuscript would need to provide some additional information regarding the proliferation/density/state of cells within the tissue also.

Reviewer #2:

In this study, the authors utilized their newly developed mesh-based computational framework, MOSES, to analyze the collective motion of the cells and interactions between epithelial monolayers before and after their lateral collision. They have examined three main epithelial interactions that occur in normal esophagus to esophageal adenocarcinoma progression. These interactions (Video 2) lead to formation of a stable boundary with highly dynamic motion of the two cell populations after collision (in squamous-squamous interactions), a stable boundary with less dynamic motion of cell populations after collision and having squamous monolayer pushed by columnar monolayer (squamous-columnar), and no boundary formation with retraction of squamous cell population after collision (squamous-cancer). Authors mainly define two measurements (boundary formation index and motion stability index) to quantify the differences of these boundaries. Both measurements result in selection of squamous-columnar as stable boundary while leave the other two cell combinations in the "unstable boundary" category. Further, authors investigate the effect of activation of EGFR signaling pathway in boundary formation using MOSES. While the approach is appropriate, the current level of analyses does not provide substantial enough improvement relative to other works to justify publication in *eLife*. There are a number of critiques that authors should address before considering this manuscript for publication.

1) Although the nature of boundaries between squamous-squamous and squamous-cancer are distinct, both are considered as "no boundary" based on boundary index analysis and "unstable boundary" based on stability index analysis. Authors should define new measurements to extract the differences between these two boundaries.

2) As stated by the authors, PIV can be also used for motion extraction in dense monolayers. Authors should clearly state how potential users might benefit using MOSES that otherwise would not be possible using PIV.

3) How would the mesh analysis be affected in case that there would be no continuity in monolayers after collision (i.e. occurrence of detached patches of cells after collision)? Could it be captured in disorder index analysis?

4) Authors have used no serum condition in order to disrupt the cell-cell contact within the monolayers. As serum may affect numerous cell functions, it is not clear if loss of collective sheet migration is because of the reduced cell-cell contacts. Authors are encouraged to use cells that have deleted intercellular adhesion (e.g. α catenin).

5) It would be beneficial to measure migratory behavior of each superpixel with respect to its distance from the boundary before and after collision.

6) How may the analyses change in cases that a combination of stable and unstable boundaries exist in the same field of view?

7) Authors state that in squamous-squamous interactions when cells are exposed to 20 ng/ml of EGF the values of boundary and stability indices are similar to that of squamous-cancer interactions. However, there is no retraction observed in squamous-squamous interactions (Video 7). Again, this needs to be resolved.

Reviewer #3:

The manuscript under review describes a method (MOSES) to quantify cell dynamics. The method is tested with epithelial monolayers made of different cell types. MOSES appears to be an interesting approach to the quantification of cell dynamics. However, for the reasons outlined below, I cannot recommend publication of the manuscript in the present form.

1) The paper is too technical to be really useful for potential users. In particular:

1a) The description of the track filtering and mesh formulation steps are so involved that in the end, I did not understand how a mesh is generated and used.

1b) In a similar manner, the boundary formation index (and to some extent also the motion stability index) is introduced in a way that requires a bona fide effort from the reader to believe that it really measures what is claimed. In my view, a methodological paper like this one, does not benefit from having all the important definitions in the Materials and methods section. Also, it would be greatly appreciated to be able to understand why the proposed parameters for the quantification of the cell dynamics outperform other possible choices.

1c) There are parts that are incomprehensible, such as for instance the paragraph entitled Automated cell counting with convolutional neural networks. This is a pity because the principal component analysis represents one of the most promising features of the method.

1d) Some quantities are not sufficiently defined. For instance, how is TrackMate similarity in Figure 2—figure supplement 2C defined? How is the distance defined in the vertical axis of Figure 1 - figure supplement 5B?

1e) Some choices are not clearly explained: why the cut-off for boundary formation and stability are defined by using the standard deviation? This choice seems to me to be an ambiguous one. For instance, a cut-off for boundary formation is "defined statistically as one standard deviation *higher* than the pooled mean of all three combinations (Figure 3C). Above this cut-off, cell combinations are categorised as forming a boundary". Surprisingly, a few lines below the authors write "Similarly, experiments in 0% serum were used to set the global motion stability threshold (0.87), one standard deviation *below* the pooled mean" (Figure 3D). Can the authors explain this asymmetry? The use of standard deviations for fixing thresholds would need in my view that a band exists (pooled mean +/- standard deviation) where the behavior is not determined. Only below mean + sd and above mean - sd a clear behavior could be attributed.

I suggest that the authors consider seriously revising the structure of their paper to make it understandable to a wide audience and to enable the potential reader to properly evaluate the powerfulness and the limitations of the proposed approach.

2) The literature review presented in the Introduction is also not clear, being a puzzling mix of methods to quantify the cell dynamics (such as tracking and PIV) and models (such as the vertex model). To me, the authors fail in merging successfully these two branches of literature and, in particular, in explaining where does MOSES fit and why the previously available tools, are not as good as MOSES in terms of robustness, sensitivity, automatization, and unbiasedness.

3) There are some claims that seem not to be supported by the experimental results. Typical examples are:

3a) Subsection “In-vitro model to study the spatio-temporal dynamics of boundary formation between different cell populations”, first paragraph: Figure 1—figure supplement 2 is cited in support of the fact that cells labeled with different color dye move in a similar way. Inspection of SF2(Figure 1 - figure supplement 2) shows that in some cases the MSD can differ by about one order of magnitude. How is it that the authors consider this a proof of similarity?

3b) The method is allegedly working perfectly in an automatic and unbiased fashion. How comes that the boundary in Figure 3—figure supplement 9A does not seem to be adequately determined?

3c) The statement "The mesh disorder index showed statistically significant increases with EGF concentration" does not seem to be supported by the data in Supplementary Figure 9, where a non-monotonic behavior would also be compatible with the data.

4) There are some claims whose general validity (i.e. in other experiments) is rather dubious.

4a) The Authors claim that "individual cells behave similarly to their neighbors so global motion patterns can be used as a proxy to study single cell behavior". I doubt that this is always true. For instance, for the liquid states found in [Nature Materials 16, 587-596 (2017)] I believe that this claim wouldn't be true.

4b) Given that the L1-norm at is not defined, what is the general validity of the statement "We use L1 for robustness as this value is > 1"?

5) The authors propose MOSES as a robust, sensitive, automatic and unbiased method and I trust them that this might be true. However, the current version of the manuscript proves that MOSES works fairly well in the few cases selected by the authors and it is not clear to me that it could be really used successfully in all other cases. Maybe, the authors could comment on this by suggesting cases in which they expect MOSES to work and cases where they think it would not. For instance, do the authors think that MOSES would work for the experiments in [Nature Materials 16, 1029 (2017)]?

In summary, I do see some potential in MOSES but the current version is not making a good job in explaining clearly what the method does, why is it better than other approaches and when it should be applicable. I hope that the authors will be able to address the above issues.

[Editors’ note: what now follows is the decision letter after the authors submitted for further consideration.]

Thank you for submitting your article "Motion Sensing Superpixels (MOSES): A systematic framework to quantify and discover cellular motion phenotypes" for consideration by *eLife*. Your article has been reviewed by three peer reviewers, and the evaluation has been overseen by a Reviewing Editor and Didier Stainier as the Senior Editor. The following individuals involved in review of your submission have agreed to reveal their identity: Roberto Cerbino (Reviewer #2); Alison McGuigan (Reviewer #3).

The reviewers have discussed the reviews with one another and the Reviewing Editor has drafted this decision to help you prepare a revised submission.

This manuscript is well improved and of high scientific value; however two of the reviewers raised several significant points that need clarification. These should be addressed in detail to resolve their concerns. The issue of serum-free condition as negative control may require additional experimentation for validation.

Reviewer #1:

The second version of the manuscript has been improved extensively and reads better than the first one. However, there are a few criticisms the authors should address before considering this manuscript for publication.

An important feature of the tissue boundaries is their shape. It would be beneficial if authors could also incorporate the analysis of boundary roughness in their computational framework.

Authors stated that the "cancer cell line OE33 pushed EPC2 out of the field of view". Since the traction forces applied at the interface between the two sheets is not presented in the manuscript, it is unclear whether the EPC2 cells retracted upon contact with OE33 cells or are continuously pushed by application of physical forces exerted by OE33 cells at the interface.

Authors have conducted new experiments, where the impact of depletion of Ca^2+^ on boundary formation is examined, to elucidate how the absent of cell-cell contact may disrupt the collective motion of the epithelial sheets. However, it is still unclear how serum depletion impacts the proliferation rate of the cells and that affects the collective motion of the sheets.

In conclusion, authors should present how much their computational framework can potentially improve our knowledge in biological processes and how it could be used to tackle new challenges.

Reviewer #2:

The authors have done an extensive amount of work to try to address all the reviewers' comments. As a result, the manuscript is now more clear and convincing in describing the proposed MOSES approach for quantifying the motility of cells belonging to a collective. I thus recommend publication in *eLife*.

Reviewer #3:

This paper presents a potentially useful tool to enable quantification of cell movement from large numbers of movies to better identify molecules that modulate collective cell migration dynamics. The scale of the data collected is impressive but I found this paper incredibly challenging to read and to understand what the data measurements physically represented and what these measurements meant biologically for our understanding of cell cooperation during boundary formation. Furthermore, I did not really understand the argument made by the authors that serum free represents no boundary formation versus for example delayed boundary formation since cells will move slower and proliferate less (and have a gap to fill before a boundary can form). This manuscript is interesting but is not currently accessible to a broad readership in my opinion.

The major comment I had that I think could significantly improve the impact of the work is can the authors better highlight the functional importance of the metrics they are quantifying in terms of the biological behaviours. For example is a boundary the best thing to be describing here or is wound healing a better example to be able to extract out different cell movement regimes to highlight the power of the tool? I could not understand how the metrics highlighted here could give me new insight into how a boundary is formed therefore it was not clear what I could learn using the tool.

The use of serum free medium to prevent boundary formation does not seem the most robust approach as this also likely impacts cell movement and proliferation, which will impact the timing required to form the boundary in the model being used (since the cells have to proliferate to fill the open space between the two cell domains. I am not sure of the logic behind specifying that doing things in serum free media gives a negative control that corresponds to a no-boundary formation case.

---

## [Author Response]

[Editors’ note: the author responses to the first round of peer review follow.]

Reviewer #1:In the manuscript "Motion Sensing Superpixels (MOSES): A systematic framework to quantify and discover cellular motion phenotypes", Zhou et al. present a new method for measuring cell motility in epithelial sheets and apply this to boundary formation in Barrett's Esophagus.The approach presented fills the technical gap between PIV/Optical flow measurements and single-cell tracking and provides a method to characterise, and through PCA, discover, motion phenotypes within 2D epithelia. This is certainly a technically interesting concept that is relatively simple to implement and provides insight into motion phenotypes within a tissue. This method will be broadly applicable to many problems in characterising cellular motion. In general I find the manuscript well written and comprehensive in its description of the technique, but not particularly easy to read.One of the difficulties is that it is unclear to me whether the authors intend to reveal new insight into the fundamental phenomena or simply provide a framework for automating the analysis for e.g. drug discovery.

MOSES is a computational framework that was developed with an aim to reveal new insights into fundamental phenomena of cellular motion dynamics. However, the biological data generated in this study are best suited to demonstrate the potential of MOSES rather than to conclude any new insight into the fundamental biological principles of cellular motion dynamics. In the revised manuscript, we have carried out a comprehensive comparison between MOSES and a widely used biological motion analysis framework, PIV (Particle Image Velocimetry). We illustrated a number of advanced features of MOSES over PIV and additionally have now emphasised the potential of MOSES in automated high-throughput screening applications. The ability of MOSES to employ a systematic analysis pipeline to generate motion maps in an unbiased way that captures local and global motion patterns through a dynamic mesh construction, will enable future study of novel biological insights in cellular motion dynamics. These points have now been clarified in the revised manuscript.

I would argue that cell state (sub cellular protein levels, cell cycle, cell death/extrusion, as examined in Schmitz et al., 2010, Pau et al., 2013, Held et al., 2010) are equally important and are not addressed by the current method.

General methods for quantifying cell states are indeed important and pattern recognition methods are being applied to analyse protein levels. When cells are sparse, existing published software such as TrackMate, CellProfiler, CIV can be used for single cell or particle tracking. Our work here, however, primarily focuses on the extraction and analysis of collective biological motion and complex phenotypes regardless of cellular density for which software is rarely available and current analysis capabilities are limited. We have now mentioned this in the Discussion.

In the present manuscript, only the cell motion is considered and yet cell proliferation/density is likely to be very important also. Can the authors provide cell density measurements (a proxy for cell proliferation) as a function of time, perhaps using the CNN counting approach, to give some estimate of motion versus proliferation?This would be particularly interesting in light of the experiments involving titration of EGF. These are very interesting and provide some insight into how collective dynamics/cell adhesion are important in the tissue mechanics in general.

Thank you for the suggestion. We considered that for cell counting to accurately reflect cell proliferation rate, the whole field of view should be covered or the cell population should be largely static (not migratory). In our videos of migrating epithelial sheets, the cells are both moving and proliferating. In these circumstances we propose two approaches to estimate cell proliferation independent of migration.

i) Cell density counting using CNNs (Materials and methods). To avoid issues of moving areas, cells were counted from randomly sampled equal sized image window patches of 64x64 pixels with full cell coverage inside. The average cell density for each cell population was reported as the average over 100 random such windows per time point (Materials and methods).

ii) Quantification of the rate in fluorescence decay. As fluorescently stained cells divide, over time the dye intensity exponentially decays. The faster the division rate, the faster the decay rate. (Materials and methods).

As requested, we applied this to the EGF experiments. Results are shown in new Figure 4—figure supplement 1. Both methods show no differences in the cell proliferation rates of a confluent epithelial sheet with increasing EGF concentration in 5% serum. This is different from a sparse culture in which EGF concentration is more likely to influence cell proliferation rate.

Another consideration is that of axial/upward motion at the interface between the two epithelial monolayers as they meet. Presumably the MOSES approach would register very little motion in a region where cells are being extruded upward (as in Figure 1B). How do the authors envisage addressing this?

Our primary interest in developing a scalable approach for high-throughput screening led us to focus our work on monolayers of cells and 2D timelapse videos, which are the most readily available. However we note that Boquet-Pujadas et al. (BioFlow: a non-invasive, image-based method to measure speed, pressure and forces inside living cells. Scientific reports 7.1 (2017): 9178) have shown already that the optical flow method of motion extraction can be extended with additional mathematical constraints to capture out-of-plane motion. Although the current version of MOSES does not contain such a feature however it can be extended and applied on 3D+time imaging to capture such extrusion processes in the future. This possibility has been discussed in our revised manuscript (see revised main text).

Reviewer #2:[…] 1) Although the nature of boundaries between squamous-squamous and squamous-cancer are distinct, both are considered as "no boundary" based on boundary index analysis and "unstable boundary" based on stability index analysis. Authors should define new measurements to extract the differences between these two boundaries.

To address this reviewer’s comment, we quantified the observed differences between the squamous-squamous and squamous-cancer boundaries displaced distance and the results is now shown in the new Figure 1G.

2) As stated by the authors, PIV can be also used for motion extraction in dense monolayers. Authors should clearly state how potential users might benefit using MOSES that otherwise would not be possible using PIV.

We include a new section and a summary table in the revised manuscript that explicitly compares the benefits of MOSES with respect to PIV using two published datasets. Please see Results section “Comparison between MOSES and PIV” and new Figure 6.

3) How would the mesh analysis be affected in case that there would be no continuity in monolayers after collision (i.e. occurrence of detached patches of cells after collision)? Could it be captured in disorder index analysis?

To address this reviewer’s comments to perform the mesh analysis when there is no continuity in monolayers after collision we used a video of EPC2 (R):OE33 (G) cells with 5ng/ml of Lapatnib (EGF inhibitor). At this concentration, after the two sheets collide, EPC2 cells rapidly die and shrivel up, creating numerous ‘gaps’ in the epithelial sheet. The results are shown in the new Figure 3—figure supplement 3, where these irregular movements in EPC2 are clearly captured in snapshots of the mesh (Figure 3—figure supplement 3A), and registered in the corresponding mesh strain curve (Figure 3—figure supplement 3B).

4) Authors have used no serum condition in order to disrupt the cell-cell contact within the monolayers. As serum may affect numerous cell functions, it is not clear if loss of collective sheet migration is because of the reduced cell-cell contacts. Authors are encouraged to use cells that have deleted intercellular adhesion (e.g. α catenin).

Although we agree with this reviewer that depleting α catenin will disrupt cell/cell contact, to be able to disrupt and re-establish cell-cell contacts with high efficiency and reproducibility is crucial to test the robustness of MOSES. This is the reason why we did not use genetic manipulation to disrupt cell-cell contacts because the efficacy of RNAi or CRISPR mediated depletion of any given gene can be highly variable among different experimental conditions and it is not easily reversible.

Depletion of Ca^2+^ is very well known to disrupt the cell-cell contacts in epithelial cells and addition of Ca+ can also induce the re-establishment of the cell/cell contacts. Thus we measured the corresponding boundary formation index in cells with Ca+ depletion or with various concentrations of Ca+ added back to culture mediums as indicated. Additionally we performed the experiment with different FBS concentrations and different media; KSFM, KSFM+RPMI and KSFM+RPMI+FBS, to control for potential media effects. To measure motion collectiveness we used the standard velocity order parameter. The results are shown in the Author response image 1. Across all samples a significant correlation between the motion collectiveness as measured by the velocity order parameter and the boundary formation index (Pearson correlation coefficient, r = 0.41) was found when all n = 33 samples were pooled together (right hand scatter plot in Author response image 1).

All numerical concentrations without units in the figure refer to ng/ml, thus Cal:1000 denotes a 1000ng/ml concentration of Ca^2+^.

5) It would be beneficial to measure migratory behavior of each superpixel with respect to its distance from the boundary before and after collision.

Due to the difficulty of representing this information for all the superpixels (1000) across 144 frames and in 2D with a moving boundary, we approximated the interface between the two cells as a vertical line and constructed velocity kymographs from the extracted dense optical flow which the superpixels use to update their (*x*,*y*) positions, as in the paper of Trepat (Rodríguez-Franco et al., 2017). These are included as the new Figure 3A for the three principal cell combinations EPC2:EPC2, EPC2:CP-A and EPC2:OE33 in 0% and 5% serum, and as the new Figure 4B for EGF addition to EPC2:CP-A in serum.

6) How may the analyses change in cases that a combination of stable and unstable boundaries exist in the same field of view?

Unfortunately we do not have a video dataset where we have exclusively studied the coexistence of stable and unstable boundaries in the same field of view. However our experiments with EGF inhibition when applied to the cell combinations of EPC2:CP-A (squamous-columnar) and EPC2:OE33 (squamous-cancer) contain a small set of examples for illustration of the required changes and we report our results in Author response image 2.

In Author response image 2, we observed partial rescue of the EPC2:OE33 phenotype upon EGF inhibition with motion behaviour more similar to EPC2:CP-A however at these concentrations the EGF inhibitor (Lapatnib) is toxic to the cells and leads to death. This effectively induces the formation of ‘multiple unstable boundaries’ in the EPC2 cells with a moving OE33 front. The boundary formation index cut-offs established in the paper in new Figure 3C implicitly assume the presence of one boundary. Applied here, whilst the boundary formation index still captures the motion concentration in the motion saliency map, as stated in the new main text, it will no longer be specific in only quantifying the interface (the boundary) between the two cell populations (Author response image 2). To handle the presence of multiple boundaries, one can segment the different unique boundaries by thresholding on the motion saliency map and deriving more specific measures such as counting the number of ‘spot-like’ sites.

**Author response image 2. respfig2:** 

7) Authors state that in squamous-squamous interactions when cells are exposed to 20 ng/ml of EGF the values of boundary and stability indices are similar to that of squamous-cancer interactions. However, there is no retraction observed in squamous-squamous interactions (Video 7). Again, this needs to be resolved.

We assume by retraction the reviewer is referring to the squamous-columnar (EPC2:CP-A) combination where the CP-A cells push the squamous EPC2 cells out of the field of view. We have quantified this displaced distance relative to the size of the image in the new Figure 4D with the EPC2:OE33 videos as reference. Grouping the displaced distance in terms of <5ng/ml and >=5ng/ml, the difference in displaced distance is highly significant, and there was no statistically significant difference in the displaced distance between EPC2:CP-A with 20ng/ml EGF and EPC2:OE33.

Reviewer #3:The manuscript under review describes a method (MOSES) to quantify cell dynamics. The method is tested with epithelial monolayers made of different cell types. MOSES appears to be an interesting approach to the quantification of cell dynamics. However, for the reasons outlined below, I cannot recommend publication of the manuscript in the present form.1) The paper is too technical to be really useful for potential users. In particular:1a) The description of the track filtering and mesh formulation steps are so involved that in the end, I did not understand how a mesh is generated and used.

We have simplified the definitions and abstracted the details more in the main text and the Materials and methods section, and refer interested/more specialised users to the Materials and methods and source code. We have added a schematic diagram of MOSES as a new Figure 2, and the new Figure 2—figure supplement 1 illustrates the technical explanation of the track filtering, which is detailed in the Materials and methods.

Track Filtering:

Our regular spaced superpixels across the whole image act like mini-cameras that assess the local velocity. In the initial frames, only those superpixels that cover the epithelial sheet will move. The rest will remain static as they cover the background image pixels that are black. The objective of the track filtering is to only identify those ‘wanted’ superpixels that cover the movement of the epithelial sheet. Using the above observation, briefly the wanted superpixels can be captured by thresholding on the observed velocities of superpixels in the initial few frames. We have added new text on this in the Materials and methods section of the main text.

Mesh formulation:

The mesh formulation aims to capture the movement of individual superpixels with respect to its local neighbouring superpixels. This involves taking each individual superpixels and connecting it to its neighbours. What constitutes a neighbour? Mathematically this is user defined based on the particular application. For example the nearest k superpixels based on distance gives rise to the popular k-NN neighbour graph when considering flocking behaviour, or one can choose all superpixels within a cutoff distance d_c_ as we have done here. The definition of neighbour can be recalculated at every time frame but here we only compute the neighbour at the initial timepoint, frame = 0 and assume the same neighbours are retained in all subsequent frames. By doing so if the distance between the neighbours increases dramatically this suggests disruptive motion (a group of cells that started close do not end up in the same spatial area). We have added new text also on this in the Materials and methods section of the main text.

1b) In a similar manner, the boundary formation index (and to some extent also the motion stability index) is introduced in a way that requires a bona fide effort from the reader to believe that it really measures what is claimed.

We apologise for being too complex in our descriptions, and have worked to improve them throughout the manuscript. The boundary formation index is derived from the motion saliency map which is theoretically motivated by Lagrangian mechanics (subsection “Quantitative measurement of

squamous and columnar epithelial boundary formation using MOSES”, second paragraph) in the new main text and above discussion regarding divergence of a motion field).

The mesh stability index (called motion stability index in the original manuscript) is introduced to measure the rate of movement of neighbouring superpixels relative to each other. If the distances between adjacent superpixels are constantly moving then this leads to an unstable mesh and a low stability index. On the other hand if the distance between adjacent cell groups do not change with respect to their initial configuration then the stability index is high. We highlight this property with simulated data (see new Figure 3—figure supplement 4), and have endeavoured to make all the descriptions easier for readers outside the field to follow.

In my view, a methodological paper like this one, does not benefit from having all the important definitions in the Materials and methods section. Also, it would be greatly appreciated to be able to understand why the proposed parameters for the quantification of the cell dynamics outperform other possible choices.

We have now ensured that the important definitions are explained in the main text and refer only to the very detailed definitions and methods in the supplementary material. We also now include a table (new Table 1) whereby each statistical indices in the paper has a biological definition/interpretation (e.g., mesh order measures the collectiveness of local cellular migration) and example biological applications. Finally a number of new figures have been introduced to illustrate these more complicated concepts (see new Figure 3—figure supplements 2-5).

1c) There are parts that are incomprehensible, such as for instance the paragraph entitled Automated cell counting with convolutional neural networks. This is a pity because the principal component analysis represents one of the most promising features of the method.

We apologise for the confusion. The paragraph entitled ‘automated cell counting with convolutional neural networks’ in the Materials and methods has now been edited for clarity. A new supplementary figure has been added (see new Figure 1—figure supplement 1) that clarifies better the training and application procedure of the convolutional neural network.

1d) Some quantities are not sufficiently defined. For instance, how is TrackMate similarity in Figure 2—figure supplement 2C defined? How is the distance defined in the vertical axis of Supplementary Figure 9D?

Similarity of MOSES tracks with TrackMate tracks are measured using the normalised cross-correlation between the paired tracks (see new Figure 2—figure supplement 2 legend). The potentially confusing original Supplementary Figure 9 showing the mesh disorder index is now removed and no longer part of the manuscript. The distance was in number of superpixels as in the spatial correlation computation of Figure 1—figure supplement 5B.

1e) Some choices are not clearly explained: why the cut-off for boundary formation and stability are defined by using the standard deviation? This choice seems to me to be an ambiguous one.

The metrics exhibit a unimodal distribution (see violin plots of new Figure 3C-E). Correspondingly it is statistically justified to use the standard deviation and the mean to set cut-offs. This practice is equivalent to using a t-distribution/normal distribution and in line with the standard practice of one-tailed t-tests. We have added a new section in the Materials and methods to explain this more clearly.

For instance, a cut-off for boundary formation is "defined statistically as one standard deviation higher than the pooled mean of all three combinations (Figure 3B). Above this cut-off, cell combinations are categorised as forming a boundary".

The higher the boundary formation index, the greater the probability of forming a boundary. Thus we set one standard deviation higher than the pooled mean. This corresponds to a one-tailed t-test where we need to observe values at least as high as this threshold to confidently predict a boundary. The null hypothesis is the absence or insufficient evidence for boundary formation with the pooled combinations.

Surprisingly, a few lines below the authors write "Similarly, experiments in 0% serum were used to set the global motion stability threshold (0.87), one standard deviation below the pooled mean". Can the authors explain this asymmetry?

Similarly the lower the mesh stability the more the epithelial sheet moves as a whole. We set one standard deviation lower than the pooled mean corresponding to the one-tailed test where we need to observe values at least as low as this to predict instability. The null hypothesis is that with the pooled combinations we already see mesh stability. Therefore the interesting phenomena to test for here is instability.

By choosing the cut-offs in this way we are being more stringent with what constitutes ‘boundary-forming’ and ‘unstable’ which are the two ‘interesting’ phenotypes contrary to the norm.

The use of standard deviations for fixing thresholds would need in my view that a band exists (pooled mean +/- standard deviation) where the behavior is not determined. Only below mean+ sd and above mean-sd a clear behavior could be attributed.

Our threshold choices are equivalent to fitting the data with a 2-class classifier. The reviewer’s suggestion of +/- cutoffs corresponds to a 3-class classifier is also a plausible model where the extra class is used as a ‘miscellaneous’ class where we are unsure of the phenotype. We believe that both models are justified. We chose to use the former which is more fitting in the context of drug screening.

I suggest that the authors consider seriously revising the structure of their paper to make it understandable to a wide audience and to enable the potential reader to properly evaluate the powerfulness and the limitations of the proposed approach.

We have improved the readability of the manuscript during our revision and the revised manuscript have been read by various colleagues including a former journal editor outside the field. We hope the revised manuscript is now more understandable to the broader reader.

2) The literature review presented in the Introduction is also not clear, being a puzzling mix of methods to quantify the cell dynamics (such as tracking and PIV) and models (such as the vertex model). To me, the authors fail in merging successfully these two branches of literature and, in particular, in explaining where does MOSES fit and why the previously available tools, are not as good as MOSES in terms of robustness, sensitivity, automatization, and unbiasedness.

We have carried out extensive modifications to the Introduction by providing more technical context for our work with respect to the existing literature. We have also included a new comprehensive comparison between MOSES and PIV, the most commonly used method and demonstrate the more advanced features of MOSES over PIV in the main text and Discussion.

3) There are some claims that seem not to be supported by the experimental results. Typical examples are:3a) Subsection “In-vitro model to study the spatio-temporal dynamics of boundary formation between different cell populations”, first paragraph: Figure 1—figure supplement 2 is cited in support of the fact that cells labeled with different color dye move in a similar way. Inspection of SF2 (Figure 1—figure supplement 2) shows that in some cases the MSD can differ by about one order of magnitude. How is it that the authors consider this a proof of similarity?

Thank you for the comments. We have investigated the issue and found a batch effect inherent in the MSD calculation. In one batch of videos, one cell type was plated more asymmetrically (occupying ~70% of the image) as opposed to the normal 50% and we didn’t filter these tracks out in the MSD computation. The corrected quantification is presented in the new Figure 1—figure supplement 2 and statistical tests have been carried out on the extracted MSD components.

3b) The method is allegedly working perfectly in an automatic and unbiased fashion. How comes that the boundary in Figure 3—figure supplement 9A does not seem to be adequately determined?

Figure 3—figure supplement 9A illustrates the process by which we infer the gap closure frame starting from image segmentation of the fluorescent images. The objective was not to delineate perfectly the sheet boundaries but to accurately infer the time of gap closure for downstream analysis. Comparison of the inferred gap closure times over n = 246 videos with the corresponding consensus manual annotations of 3 independent researchers demonstrate high correlation (0.902 Pearson correlation) and an agreement of 94% within 5 frames, (new Figure 3—figure supplement 9E) which is very comparable to the 97% accuracy between individual humans. This justifies the initial quality of image segmentation used for the intended purpose of inferring the frame in which the gap between the two sheets are closed.

3c) The statement "The mesh disorder index showed statistically significant increases with EGF concentration" does not seem to be supported by the data in Supplementary Figure 9, where a non-monotonic behavior would also be compatible with the data.

During the revision, we carried out extensive testing using our own and published datasets and concluded that the newly proposed mesh order in the revised manuscript is more general than the initially proposed mesh order index as a measure of collective motion based on our MOSES mesh construction. The previously proposed mesh disorder index was only applicable between videos with identical initial configuration i.e. same type of cells and plating as in the case of the EGF titration videos. The results shown in Supplementary Figure 9 (original manuscript) was intended to highlight that the mesh disorder index deviates from uniformity when EGF concentrations > 5ng/ml. Statistical tests were carried out to identify this deviation from a constant line model which is reasonable for 0-5ng/ml. Thus overall the behaviour is non-monotonic. With our new improved mesh order measurement, Supplementary Figure 9 of the original manuscript is now removed as we no longer use the mesh disorder index.

4) There are some claims whose general validity (i.e. in other experiments) is rather dubious.4a) The Authors claim that "individual cells behave similarly to their neighbors so global motion patterns can be used as a proxy to study single cell behavior". I doubt that this is always true. For instance, for the liquid states found in [Nature Materials 16, 587-596 (2017)] I believe that this claim wouldn't be true.

We address this point together with point 5 (see below).

4b) Given that the L1-norm at is not defined, what is the general validity of the statement "We use L1 for robustness as this value is > 1"?

We apologise for neglecting to define this norm in our previous version of the manuscript. We have added a new section “Normalised Mesh Strain, L1-norm and robustness” in the Materials and methods explaining our design choice.

5) The authors propose MOSES as a robust, sensitive, automatic and unbiased method and I trust them that this might be true. However, the current version of the manuscript proves that MOSES works fairly well in the few cases selected by the authors and it is not clear to me that it could be really used successfully in all other cases. Maybe, the authors could comment on this by suggesting cases in which they expect MOSES to work and cases where they think it would not. For instance, do the authors think that MOSES would work for the experiments in [Nature Materials 16, 1029 (2017)]?

We thank the reviewer for their help in making available the raw videos of [Nature Materials 16, 587-596 (2017)] available. Together with the videos in [Nature Materials 16, 1029 (2017)] as our validation datasets we have refined our claims in the revised manuscript. In addition we profiled the similarity and differences between MOSES and PIV. The new results are included in the revised manuscript as a new section with an additional Figure 6 and accompanying summary table.

[Editors' note: the author responses to the re-review follow.]

Reviewer #1:The second version of the manuscript has been improved extensively and reads better than the first one. However, there are a few criticisms the authors should address before considering this manuscript for publication.An important feature of the tissue boundaries is their shape. It would be beneficial if authors could also incorporate the analysis of boundary roughness in their computational framework.

We quantified the ‘boundary roughness’ according to the deviation of the geometrical boundary shape from a straight line, *L*/*L*_0_ where *L* is the length of the boundary and *L*_0_ is the length of the straight line joining the endpoints of the boundary as in Javaherian, Sahar, et al (Modulation of cellular polarization and migration by ephrin/Eph signal-mediated boundary formation." *Integrative Biology* 9.12 (2017): 934-946), (Figure 3—figure supplement 10). No significant departures from a straight line (*L*/*L*_0_=1) were found. A new paragraph for extracting the boundary to calculate boundary shape has been added to the Materials and methods.

Authors stated that the "cancer cell line OE33 pushed EPC2 out of the field of view". Since the traction forces applied at the interface between the two sheets is not presented in the manuscript, it is unclear whether the EPC2 cells retracted upon contact with OE33 cells or are continuously pushed by application of physical forces exerted by OE33 cells at the interface.

We acknowledge that without traction force measurements we can’t draw a clear conclusion about the forces at play in our assay, whether it was EPC2 cells which retracted or was continuously pushed by OE33 cells. The wording in our previous manuscript, “cancer cell line OE33 pushed EPC2 out of the field of view”, was used to describe the phenomenon of OE33 cells expanding into EPC2 cells resulting in their eventual disappearance from the field-of-view. This is supported by the derived motion field (Figure 1—figure supplement 3). Just before gap closure, EPC2 and OE33 cells are moving in opposite directions. After gap closure EPC2 cells near the interface continue to move in opposite directions to OE33 cells initially before they appear to be dominated by OE33 cells. This is suggestive of ‘pushing’. Unfortunately due to the large fluorescence decay in OE33 cells near the interface it is not fully conclusive whether this ‘pushing’ is continuously maintained for the full duration of the timelapse video. Confocal image of cell junctions in fixed samples at 72 h together with evidence of motion suggest continual application of physical forces. However further experimentation is required to confirm this under timelapse microscopy study. In the main we now state “in the squamous-cancer EPC2:OE33 combination, the cancer cell line OE33 expanded continuously, resulting in the disappearance of EPC2 from the field of view (Video 2, Figure 1E) as assessed by the motion field and confocal images (Figure 1—figure supplement 3). The forces that govern the behaviour of the two cell lines on contact are unknown and traction force microscopy is required to investigate the ‘retracting’ or ‘pushing’ behaviour of EPC2 or OE33 cells respectively in future studies”.

Authors have conducted new experiments, where the impact of depletion of Ca^2+^ on boundary formation is examined, to elucidate how the absence of cell-cell contact may disrupt the collective motion of the epithelial sheets. However, it is still unclear how serum depletion impacts the proliferation rate of the cells and that affects the collective motion of the sheets.

To address this question, we used cell density as an indicator of cell proliferation. The mean cell density and mean change in cell density over the first 48 h (≈ 16 h before and 32 h after gap closure) were estimated from cells grown in serum (5% FBS) and no serum (0% FBS) conditions from individual video frames using the established convolutional neural network approach (Materials and methods). Our results, shown in the new Figure 1—figure supplement 4, suggest that under the experimental conditions used in these studies, the lack of serum has undetectable impact on cell density. This is in complete contrast to the observed increase in collective cell migration and presence of boundary formation in 5% serum. Together these results illustrate that serum-free medium has a profound impact on cell motion dynamics. We therefore used 0% serum in the paper as a computational negative control for the detection of boundary formation. We have modified wording in the text to provide explicit clarification to avoid confusion (see subsection “In-vitro model to study the spatio-temporal dynamics of boundary formation between different cell populations”, last paragraph).

In conclusion, authors should present how much their computational framework can potentially improve our knowledge in biological processes and how it could be used to tackle new challenges.

We thank the reviewer for the constructive comment. To address this we have rewritten the Introduction to provide additional rationale for the need of developing a computational framework like MOSES. In particular we highlight the advantage of MOSES over single-cell tracking and existing PIV/CIV type methods, “we developed Motion Sensing Superpixels (MOSES), a computational framework that aims to provide a flexible and general approach for biological motion extraction, characterisation and phenotyping. We empowered PIV-type methods with a mesh formulation that enables systematic measurement and unbiased extraction of rich motion features for single and collective cell motion suitable for high-throughput phenotypic screens.”

In the revised Discussion, the potential of MOSES to advance biological knowledge and its ability to tackle new challenges is stated as the following.

1) Single-cell tracks are notoriously problematic over long times; the track of a single cell may be lost or broken into many separate tracks. MOSES superpixel tracks avoids this and recovers the global motion patterns (c.f. motion saliency maps, derived measures and motion signatures). Whilst side-by-side comparison of MOSES and the standard PIV method using published datasets demonstrates that MOSES not only enables all the measurements of PIV, but by further exploiting long-time tracks and neighbourhood relationships, delivers greater physical and biological insights. (Discussion, second paragraph).

2) Complex salient spatio-temporal motion patterns and events such as boundary formation, deformation waves due to cell jamming between two cell populations and cell death can all be quantitatively captured by MOSES. Critically, the ability of MOSES to perform long-time tracking (up to 6 days demonstrated in this study) enabled spatial localisation of the cell populations involved in a particular motion phenotype. (Discussion, second paragraph).

3) MOSES does not require complex user settings to facilitate reproducibility in analyses because it does not aim to threshold or cluster out the moving objects or phenotypes during analysis, which would introduce intermediate processing errors. Rather its philosophy is to facilitate systematic generation of many motion-related measurements based on trajectory and mesh statistics sufficient for applying machine learning methods for data-driven object segmentation, video classification and phenotype detection in large video collections (e.g. Figure 5, motion map) with minimum prior information. The main parameter the user specifies is the number of initial superpixels, which determines the spatial resolution of analysis. No complicated fitting of complex models and no special hardware such as GPUs are required. (Discussion, last paragraph).

All of these illustrate the potential of MOSES as a powerful and systematic computational framework that is particularly useful for unbiased explorative high-content screening with an aim to discover fundamental principles of cellular motion dynamics in biology and to identify factors or drugs that can alter cellular motion dynamics in disease aetiology and treatment.

Reviewer #3:This paper presents a potentially useful tool to enable quantification of cell movement from large numbers of movies to better identify molecules that modulate collective cell migration dynamics. The scale of the data collected is impressive but I found this paper incredibly challenging to read and to understand what the data measurements physically represented and what these measurements meant biologically for our understanding of cell cooperation during boundary formation. Furthermore, I did not really understand the argument made by the authors that serum free represents no boundary formation versus for example delayed boundary formation since cells will move slower and proliferate less (and have a gap to fill before a boundary can form). This manuscript is interesting but is not currently accessible to a broad readership in my opinion.The major comment I had that I think could significantly improve the impact of the work is can the authors better highlight the functional importance of the metrics they are quantifying in terms of the biological behaviours. For example is a boundary the best thing to be describing here or is wound healing a better example to be able to extract out different cell movement regimes to highlight the power of the tool? I could not understand how the metrics highlighted here could give me new insight into how a boundary is formed therefore it was not clear what I could learn using the tool.

The aim of developing MOSES is as “a computational framework that aims to provide a flexible and general approach for biological motion extraction, characterisation and phenotyping. We empowered PIV-type methods with a mesh formulation that enables systematic measurement and unbiased extraction of rich motion features for single and collective cell motion suitable for high-throughput phenotypic screens.”. We have provided additional rationale for the need of developing such a computational framework in the revised Introduction. Accordingly in this paper the metrics are presented to highlight how MOSES offers a more flexible platform to define customized motion measures related to a specific biological phenomena compared to existing methods that better discriminates between the resulting motion phenotypes. Here we did not aim to specifically shed insight into how a boundary is formed. However it is important to note that further studies using our method to conduct a high-content screen following genetic or other experimental manipulations would enable us to reveal new mechanistic insights into boundary formation in an unbiased manner.

The use of serum free medium to prevent boundary formation does not seem the most robust approach as this also likely impacts cell movement and proliferation, which will impact the timing required to form the boundary in the model being used (since the cells have to proliferate to fill the open space between the two cell domains. I am not sure of the logic behind specifying that doing things in serum free media gives a negative control that corresponds to a no-boundary formation case.

This is a miscommunication on our part. In this paper our ‘negative’ control serves only to demonstrate the sensitivity of MOSES to distinguish between the clear boundary-like behaviour of EPC2:CP-A in 5% serum and the lack of in the corresponding ‘negative’ no-serum case. We do not attempt to claim any experimental conditions used in this study are biologically important in boundary formation as some of the conditions such as serum-free condition cannot be used as a rigorous biological control. Here we used serum-free medium as a condition to test the sensitivity of MOSES because it has a profound impact on cell motion dynamics with absence of boundary formation. We have now clearly stated this in the revised manuscript (subsection “*In vitro* model to study the spatio-temporal dynamics of boundary formation between different cell populations”, last paragraph, subsection “Quantitative measurement of squamous and columnar epithelial boundary formation using MOSES”, third paragraph).

To address this reviewer’s specific comments about the impact of serum on the timing and speed of boundary formation, we carried out further measurement during the revision and our results (new Figure 1—figure supplement 4, Figure 4—figure supplement 2) showed that:

1) The presence of serum causes variation in gap closure time (within +/- 5 h) but is not significant and consistent across all cell-line combinations (Figure 1—figure supplement 4A,B). This is in contrast to the notable average increased speed in serum across all cell combinations shown in Figure 3B.

2) The presence of serum does not have significant impact on the cell movement at the assay endpoint. All videos (96 h and 144 h) have the same normalized RMSD curves at late timepoints, (Figure 1—figure supplement 4B).

3) Consistent with our findings, serum has profound impact on cell movement. In 5% serum, cells moved more collectively (Figure 3F,G, Figure 1—figure supplement 5C) and moved quicker (Figure 3A,B). However increased cell movement speed alone does not contribute to boundary formation. Addition of EGF to EPC2:CP-A cells in no-serum conditions caused notable increase in cell movement speed but had minimal impact on collective migration and boundary formation (Figure 4—figure supplement 2C,G,J,K).